# Dynamics analysis of disturbance propagation in ecosystem with proportional migration based on epidemic model

**Bingbing Qian**[1]*, **Jing Hua**[2], **Xinyue Wang**[2], **Yimin Li**[2]

**1** Basic Teaching Department, Jiangsu Shipping College, Nantong, Jiangsu, China, **2** Mathematical Sciences, Jiangsu University, Zhenjiang, Jiangsu, China

* 2682091831@qq.com

## Abstract

In order to investigate the propagation dynamics of ecological disturbances within ecological networks, we conceptualize ecological disturbances as infectious diseases and employ complex network theory to analyze ecosystems. In this approach, the transmission process of ecological interference is abstracted as the spread of infectious diseases on complex networks. Based on the principles of infectious disease models, a disturbance propagation model for ecological networks is constructed and analyzed. In this paper, species within the ecosystem are abstracted as nodes in a complex network, where connections between nodes signify predator-prey relationships. When the ecosystem is under attack, a small number of species are initially disturbed, and this disturbance spreads among the ecosystem through the food chain. Given that species possess self-recovery capabilities, some species will return to a stable state over time after being disturbed. Consequently, any species within the ecosystem can be in one of three states at a given time: undisturbed, disturbed, or recovered. By establishing and analyzing the disturbance propagation dynamics, we determine the basic reproduction number and its influencing factors, and assess the stability of the disease-free equilibrium and the endemic equilibrium. The results demonstrate that when the basic reproduction number is less than 1, the system exhibits only a disease-free equilibrium, which is globally stable. When the basic reproduction number exceeds 1, an endemic equilibrium exists, the disease-free equilibrium becomes unstable, while the endemic equilibrium is globally stable. The basic reproduction number is associated with the topological structure of the food web, the probability of disturbance propagation, and the probability of species recovery. Subsequently, we validate the conclusions of the theorem using the actual food web data of 85 species from a pine forest in Otago, New Zealand. Finally, we consider protection measures for species from a human-intervention perspective, treating species protection as species immunity. Through theoretical derivation and numerical simulation, we find that the active immunization strategy is the most

**Data availability statement:** All data underlying the findings described in this study are publicly available in the Interaction Web Database at http://www.ecologia.ib.usp.br/iwdb/.

**Funding:** The author(s) received no specific funding for this work.

**Competing interests:** The authors have declared that no competing interests exist.

effective. This is because it effectively targets and protects neighbor nodes with medium and high degrees (i.e., highly connected species), thereby inhibiting the cascade of disturbance more efficiently than random or targeted strategies.

## 1. Introduction

Disturbance is a kind of natural phenomenon and ecological process which is ubiquitous in nature. Disturbance is divided into natural disturbance and human disturbance [1]. With the aggravation of human economic activities, ecologists gradually realize that natural disturbance and human disturbance play a decisive role in the development of ecosystem. Disturbance and its propagation have different effects on population evolution, biodiversity and ecosystem structure and function [2–5]. Therefore, it is of great significance to study the ecological disturbance and its propagation process in the ecosystem for the health and stability of the ecosystem. In recent years, with the rapid development of complex networks, many practical problems are abstracted as complex networks, such as computer networks, communication networks, power networks, transportation networks, financial and economic networks, ecological networks [6]. At present, the research on the application of complex network in ecosystem mainly includes the stability, invulnerability and robustness of food webs, degree distribution characteristics, the importance of individuals in the study of complex food web, and the transmission dynamics of infectious diseases in ecological network [7–10]. On ecological disturbance, many scholars at home and abroad have done a lot of research on the types and characteristics of disturbance and the impact of disturbance on ecosystem [1–5]. However, many problems about the application of complex network to the propagation of disturbance in ecosystem still need to be further studied. In this paper, disturbance is regarded as an infectious disease that can be transmitted through the food chain among species, so the disturbance propagation in the ecosystem can be abstracted as the infectious disease propagation in the complex network. The research of disturbance propagation based on infectious disease model has been applied to power grid, computer network and other practical networks [11–12]. In 2019, Q. Wu [13] studied the power grid disturbance propagation dynamics based on infectious disease model. In this paper, we also use this idea to study the propagation of ecological disturbance. There are a lot of researches on the transmission of infectious diseases on complex networks. R. M. May and A. L. Lloyd [14] gave the basic reproduction number of SIR Epidemic Model in complex network in 2001. And they draw a conclusion that the disease will break out when the threshold is exceeded. Later, J. Liu et al. [15] introduced the birth rate and mortality rate into the complex network dynamic model in the static network. In this paper, the author puts forward the empty lattice theory, that is, dead individuals produce empty lattice, and new individuals occupy empty lattice, so as to keep the topology of the whole network unchanged. By studying the SIRS Epidemic Model on the complex network, Chun-Hsien Li [16] gave the disease-free equilibrium and endemic equilibrium solutions of SIRS Epidemic Model on complex networks, and studied the range

of solutions and the global stability of solutions. When the basic reproduction number is less than 1, there is only disease-free equilibrium and global stability. When the basic reproduction number is greater than 1, there is endemic equilibrium and global stability. To study the spread and stability of infectious diseases in the ecosystem, J.W. Huo [17] applied the infectious disease model to the ecological network, studied the transmission dynamics of infectious diseases on the ecosystem, and analyzed the global stability of disease-free equilibrium and endemic equilibrium.

Before proceeding to the formal model derivation, we briefly outline the central parameters that govern the ecosystem's dynamics. The model relies on three core concepts: (1) the disturbance propagation probability ($\beta$), which quantifies the likelihood of instability spreading from a disturbed predator or prey to its neighbor, reflecting the intensity of the trophic cascade; (2) the species recovery probability ($\sigma$), which represents the ecological resilience or self-repair capacity of a species; and (3) the basic reproduction number ($\rho$), a composite threshold metric derived from these parameters. As we will demonstrate, $\rho$ serves as the fundamental determinant of global stability, predicting whether a disturbance will fade out or become endemic.

In this paper, the food network within an actual ecosystem is conceptualized as a complex network, with species represented as nodes and food chains as edges connecting these nodes. Ecological disturbances typically initiate with the perturbation of a subset of species, which subsequently triggers the spread of the disturbance through the intricate web of food chains spanning the entire ecosystem. In the context of biodiversity conservation, human interventions often involve the implementation of protection measures for specific species. In this research, such species protection efforts are analogously regarded as species immunity. Through a systematic comparison of multiple immune strategies, the most effective approach for species protection is identified. Grounded in the infectious-disease model, a disturbance propagation dynamics model for ecological networks is constructed. This model is employed to analyze the disturbance propagation threshold, equilibrium points, and their stability characteristics. Finally, the theoretical findings are rigorously verified through simulations based on actual food web data.

Despite these advances, two critical gaps remain in the current literature regarding disturbance propagation in ecosystems. First (What is known vs. Gap): While epidemic models have been widely applied to analyze stability in static networks, most ecological models assume a closed system. Few studies have adequately explored how proportional migration (the continuous immigration and emigration of species)—a ubiquitous ecological phenomenon—affects the threshold of disturbance outbreaks. Second: While the robustness of food webs has been studied, there is a lack of systematic quantitative comparison of different 'immunization' (species protection) strategies to determine which approach most effectively inhibits disturbance spread in a realistic predator-prey topology.

To bridge these gaps, this paper establishes a disturbance propagation model for ecological networks based on the SIRS infectious disease mechanism. The novelty and main contributions of this work are as follows:

Model Innovation: We construct a dynamic model of disturbance propagation that explicitly incorporates proportional migration rates (immigration and emigration) and species self-recovery. We derive the exact analytical expression for the basic reproduction number $\rho$ and determine the threshold conditions for disturbance outbreaks.

Stability Analysis: We rigorously prove the global asymptotic stability of both the disease-free equilibrium and the endemic equilibrium using Lyapunov functions and the LaSalle's invariance principle.

Strategy Optimization: We treat species protection as 'immunization' and systematically compare three strategies: Uniform Immunization, Targeted Immunization, and Active Immunization. By analyzing the changes in the propagation threshold, we identify the optimal conservation strategy.

Empirical Validation: The theoretical findings are validated using empirical data from the Otago pine forest food web (85 species), ensuring the ecological relevance of our conclusions.

The subsequent sections of this paper are organized as follows. Section 2 presents a detailed analysis of the propagation dynamics of ecological disturbances, deriving the propagation threshold and equilibrium solutions, and providing rigorous proofs of the local and global stability of the disease-free and endemic equilibrium. Section 3 conducts a

comprehensive comparative analysis of three distinct species-protection strategies, aiming to identify the most effective protection measures by comparing the magnitudes of the transmission thresholds associated with each strategy. In Section 4, an actual ecological network is selected for in-depth modeling and simulation studies to validate the theoretical conclusions. The paper concludes with a summary of the key findings and implications in the final section.

## 2. Disturbance propagation model on networks

### 2.1. Model formulation

In this paper, the disturbance between species caused by external attack on ecosystem is regarded as an infectious disease, and study the changes of species status in the ecosystem by the SIRS model (susceptible- infective-recovered-susceptible). And we analyze the actual food web of 85 species in a pine forest in Otago, New Zealand [18]. The basic idea of infectious disease model is an important basis for constructing disturbance propagation model of ecological network. When the ecosystem is attacked by external environment, the corresponding nodes of one or several species in the network will become disturbed nodes in the initial state. The disturbed species have a certain probability to restore stability and become recovery nodes. At the same time, the recovery node also has a certain chance to return to the sensitive state and receive disturbance again. We divide all species in an ecosystem into three states. The flowsheet of this model is shown in Fig 1. All species in the food web can be divided into three types: undisturbed, disturbed and recovered, which are represented by S, I and R respectively.

Before formulating the equations, we provide the ecological justification for two core modeling assumptions:

(1) Food Chains as Disturbance Pathways: In this model, the predator-prey links function as the primary channels for disturbance propagation. This is based on the ecological principle of resource dependency and energy flow. Since species rely on trophic interactions for survival, instability in one node (e.g., a sharp decline in biomass or the accumulation of toxins) inevitably impacts its neighbors through trophic cascades—either limiting resources for predators (bottom-up effects) or reducing predation pressure on prey (top-down effects).

(2) The 'Immunity' Analogy: We conceptualize 'immunization' not merely as biological immunity to disease, but broadly as anthropogenic conservation interventions. Measures such as establishing protected areas, habitat restoration, or captive breeding effectively enhance a species' resilience to environmental fluctuations. In the model, providing such protection reduces the probability of a species transitioning from a stable to a disturbed state, which is mathematically equivalent to the 'removed' or 'immunized' state in epidemiological theory.

S (undisturbed): Undisturbed state. These species are generally stable, but may be disturbed.

I (disturbed): Disturbed state. These species have been affected by disturbance and have the ability to transmit disturbance to other undisturbed species.

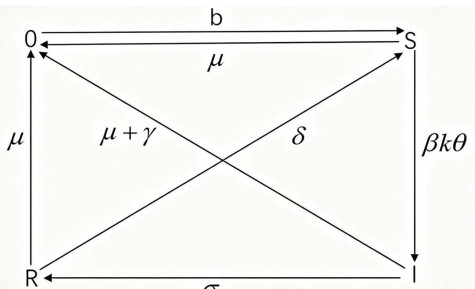

**Fig 1. The flow diagram of the SIRS model.**

R (recovered): Recovered state. This kind of species has recovered from the disturbance state, but it is still possible to become sensitive.

When a species in the food web is disturbed, the disturbance may be transmitted from the species to its predator or prey through the food chain through the food chain, making them disturbed. When the food web is regarded as a complex network, every edge of the network can be used as a way of disturbance propagation. However, even if some species have a predatory relationship with the disturbed species, the disturbance will not spread in the population because of their strong self-resistance and self-healing ability, and the species will not become disturbed. Therefore, we assume that the probability of an undisturbed species becoming a disturbed state due to its predator-prey relationship with a disturbed species is $\beta$, $0 < \beta < 1$. It is important to note that in real ecosystems, disturbances often propagate asymmetrically depending on their nature (e.g., bioaccumulation flows upwards, while trophic cascades flow downwards). However, in this theoretical framework, we conceptualize 'disturbance' as a deviation from a stable state (instability) rather than the flow of a specific physical agent. Since species in a food web are mutually dependent—predators rely on prey for energy (bottom-up), and prey rely on predators for population regulation (top-down)—a disturbance in one node has the potential to destabilize its neighbors in either direction. Therefore, to capture general topological dynamics, our model assumes that predator-prey links function as bidirectional pathways for the propagation of instability. In the same time state, the propagation probability means that if a healthy node is adjacent to one or more disturbed nodes, it will become a disturbed node according to a certain probability. Some species will be disturbed first when the ecosystem suffers from external attack. When the disturbance spreads to other species in the food web, the impact of disturbance will be reduced accordingly. With the continuous spread of disturbance and the passage of time, some less affected species will gradually recover. It is assumed that the proportion of gradually recovered species is $\sigma$, $0 < \sigma < 1$. Recovery probability means that each disturbed node becomes a recovery node according to a fixed probability at the same time. When some species are disturbed and recovered, they will have memory of the disturbance. When the disturbance comes again, they will take evasive behavior. However, some species still return to sensitive state after recovery. Therefore, we assume that the probability of the species returning from the recovered state to the undisturbed state is $\delta$, $0 < \delta < 1$. At the same time, we consider the situation that species immigration and emigration, and the species disappear in the ecosystem due to disturbance. These reasons will change the structure of the whole food web into a dynamic network. However, in order to simplify the calculation, we adopt the empty lattice theory [15], that is, the emigrated species and the species that disappear due to disturbance produce empty lattice, and the immigrated species occupy the empty lattice. Here we use 0 for the empty lattice state. We assume that the proportion of immigrated species is b, the proportion of emigrated species is $\mu$, and the proportion of species disappearing due to disturbance is $\gamma$.

For the disturbance propagation model on the actual food webs, the number of species that establish predator-prey relationship with each species is different, that is the degree of each node is different. Let $S_k(t)$, $I_k(t)$, $R_k(t)$ be the densities of undisturbed, disturbed and recovered species of nodes with scale $k$ at time $t$, respectively. Therefore, the dynamic equation of disturbance propagation can be written:

$$\begin{cases} \frac{dS_k(t)}{dt} = b\left(1 - S_k(t) - I_k(t) - R_k(t)\right) - \beta k \theta(t) S_k(t) - \mu S_k(t) + \delta R_k(t), \\ \frac{dI_k(t)}{dt} = \beta k \theta(t) S_k(t) - (\mu + \gamma) I_k(t) - \sigma I_k(t), \\ \frac{dR_k(t)}{dt} = \sigma I_k(t) - \mu R_k(t) - \delta R_k(t), \quad k = 1, \cdots, k_{\max}. \end{cases} \tag{2.1.1}$$

the probability $\theta(t)$ denotes the proportion of a species that has a direct predator-prey relationship with the disturbed species in the system. The expression of $\theta(t)$ is as follows:

$$\theta(t) = \sum_{k'} \frac{\varphi(k') P(k'|k) I_{k'}(t)}{k'}, \tag{2.1.2}$$

where the conditional probability $P(k'|k)$ is the probability that a species with node degree $k'$ has a direct predator-prey relationship with the species with node degree $k$. $P(k'|k) = k'P(k')/\langle k \rangle$, $\langle k \rangle$ is the average degree of the complex network, which represents the average value of the food chain of all species in the actual food network in this paper. $P(k')$ is the degree distribution of food network, representing the proportion of species with node degree $k'$. $\varphi(k')$ represents the propagation probability of perturbed species with node degree $k'$. In the ecological network, assume that each species has the same probability of contact with its predatory species, then the node's transmissibility of each species should be proportional to the node degree, i.e., $\varphi(k') = \alpha k'$, $0 < \alpha \leq 1$, here, we take $\alpha = 1$, that is $\varphi(k') = k'$. We clarify that $\alpha = 1$ is a modeling simplification and briefly discuss how a generalized $\varphi(k')$ would rescale or weight $\rho$ without altering its qualitative threshold behavior. We can calculate that:

$$\theta(t) = \sum_{k'} \frac{\varphi(k') P(k'|k) I_{k'}(t)}{k'} = \sum_{k'} \frac{k'P(k') I_{k'}(t)}{\langle k \rangle} = \sum_{k} \frac{kP(k) I_k(t)}{\langle k \rangle}.$$

(2.1.3)

We define the density of undisturbed species, disturbed species and recovered species in the food network as follows:

$$\begin{cases} S(t) = \sum_{k} P(k) S_k(t), \\ I(t) = \sum_{k} P(k) I_k(t), \\ R(t) = \sum_{k} P(k) R_k(t). \end{cases}$$

(2.1.4)

And the sum of species of three states corresponding to each node degree is as follows:

$$S_k(t) + I_k(t) + R_k(t) = N_k(t), k = 1, \cdots, k_{\max}.$$

(2.1.5)

## 2.2. Analysis of equilibrium solution

In this section, we reveal the properties of the solution of system (2.1.1), and calculate the disease-free equilibrium solution and endemic equilibrium solution.

**Lemma 1.** Let $(S_1(t), I_1(t), R_1(t), \cdots, S_{k_{\max}}(t), I_{k_{\max}}(t), R_{k_{\max}}(t))$ be the solution of system (2.1.1). Suppose the solution satisfies the following initial value condition: $S_k(0) \geq 0$, $I_k(0) \geq 0$, $R_k(0) \geq 0$ and $\theta(0) > 0$. Then for all $t > 0$, we have $0 < S_k(t) < 1$, $0 < I_k(t) < 1$, $0 < R_k(t) < 1$, $N_k(t) < 1$ and $\theta(t) > 0$ for $k = 1, \cdots, k_{\max}$.

**Proof.** We first prove that $\theta(t) > 0$ for all $t > 0$. From the second equation of system (2.1.1) and (2.1.3), we can calculate that:

$$\theta'(t) = \frac{1}{\langle k \rangle} \sum_{k} kP(k) I'_k(t) = \theta(t) \left( \frac{1}{\langle k \rangle} \sum_{k} k^2 \beta P(k) S_k(t) - (\mu + \gamma + \sigma) \right).$$

(2.2.1)

From (2.2.1) we can get that

$$\theta(t) = \theta(0) \exp \left[ \frac{\int_0^t \sum_k k^2 \beta P(k) S_k(\tau) d\tau}{\langle k \rangle} - (\mu + \gamma + \sigma)t \right].$$

Obviously, $\theta(t) = 0$ is a solution of (2.2.1). From the uniqueness of the solution, if $\theta(0) > 0$, we can have $\theta(t) > 0$ for all $t > 0$. By adding the three equations of system (2.1.1), we can get

$N'_k(t) = b - (b + \mu) N_k(t) - \gamma I_k(t) \le b - (b + \mu) N_k(t)$. Then we have

$$N_k(t) \le \frac{b}{(b + \mu)} + \left( N_k(0) - \frac{b}{(b + \mu)} \right) \exp\left( -(b + \mu)t \right).$$

Therefore, if $0 < N_k(0) \le b/(b + \mu)$, then $0 < N_k(t) \le b/(b + \mu) < 1$, and if $b/(b + \mu) < N_k(0) < 1$, then $0 < N_k(t) \le N_k(0) < 1$. To sum up, from $0 < N_k(0) < 1$, we have $0 < N_k(t) < 1$, $t > 0$ for $k = 1, \cdots, k_{max}$.

Because $S_k(0) \ge 0$, so if $S_k(0) > 0$, then through the continuity of $S_k(t)$, there exists a small $\varepsilon_1 > 0$ satisfied that $S_k(t) > 0$ for $t \in (0, \varepsilon_1)$, and if $S_k(0) = 0$, then

$$S'_k(0) = b(1 - N_k(0)) - \beta k\theta(0) S_k(0) - \mu S_k(0) + \delta R_k(0) = b(1 - N_k(0)) + \delta R_k(0) > 0.$$

From the monotonicity, there must exists $\varepsilon_2 > 0$, such that $S_k(t) > 0$ for all $t \in (0, \varepsilon_2)$, in summary, let $\varepsilon = \min\{\varepsilon_1, \varepsilon_2\}$, we can have $S_k(t) > 0$ for $t \in (0, \varepsilon)$.

Now we prove that $S_k(t) > 0$ for all $t > 0$. Here we use the reductio ad absurdum.

Suppose not. We can find that there exists $t_0 \ge \varepsilon > 0$ satisfied that $S_k(t_0) \le 0$, let $t_0$ be the infimum of all such points, we can get that $S_k(t) > 0$, $S_k(t_0) = 0$ for $t \in (0, t_0)$. By system (2.1.1), we can get that

$I'_k(t) + (\mu + \gamma + \sigma) I_k(t) = \beta k\theta(t) S_k(t) > 0$ for $t \in (0, t_0)$,

we can calculate that $I_k(t) > I_k(0) \exp\left( -(\mu + \gamma + \sigma) t \right) \ge 0$ for $t \in (0, t_0)$. Then

$R'_k(t) + (\delta + \mu) R_k(t) = \sigma I_k(t) > 0$ for $t \in (0, t_0)$, thus, $R_k(t) > R_k(0) \exp\left( -(\delta + \mu) t \right) \ge 0$ for $t \in (0, t_0)$.

Because of the continuity of $R_k(t)$ and $I_k(t)$ we have $R_k(t_0) \ge 0$ and $I_k(t_0) \ge 0$, Combining the first equation of system (2.1.1), $S_k(t_0) = 0$, $0 < N_k(t) < 1$ and $\theta(t) > 0$ for $k = 1, \cdots, k_{max}$, we can have that

$S'_k(t_0) = b(1 - N_k(t_0)) - \beta k\theta(t_0) S_k(t_0) - \mu S_k(t_0) + \delta R_k(t_0) = b(1 - N_k(t_0)) + \delta R_k(t_0) \ge b(1 - N_k(t_0)) > 0.$

It indicates that $S_k(t') \le 0$ for some $t' \in (0, t_0)$. Obviously, this is contradictory. Therefore, we can have $S_k(t) > 0$ for all $t > 0$. Through system (2.1.1), we can also get that $I_k(t) > 0$ for all $t > 0$ and $R_k(t) > 0$ for all $t > 0$. Because of $N_k(t) = S_k(t) + I_k(t) + R_k(t) < 1$, since $S_k(t) > 0$, $I_k(t) > 0$ and $R_k(t) > 0$, then $S_k(t) < 1$, $I_k(t) < 1$ and $R_k(t) < 1$ naturally hold. Thus, we can get $0 < S_k(t) < 1$, $0 < I_k(t) < 1$, $0 < R_k(t) < 1$, $N_k(t) < 1$ and $\theta(t) > 0$ for all $t > 0$, This completes the proof. □

**Lemma 2.** All feasible solutions of the system (2.1.1) are ultimately bounded, and the set $\Omega$ is the positive invariant set of the system, where $\Omega = \left\{ (S_k(t), I_k(t), R_k(t)) \in R_+^{3k_{max}}, k = 1, 2, ..., k_{max}, N_k(t) = S_k(t) + I_k(t) + R_k(t) \le b/(b + \mu) \right\}$.

**Proof.** First, by adding the three equations of system (2.1.1), we can get

$$\begin{aligned}
\frac{dN_k(t)}{dt} &= b(1 - S_k(t) - I_k(t) - R_k(t)) - \mu(S_k(t) + I_k(t) + R_k(t)) - \gamma I_k(t) \\
&= b - (b + \mu) N_k(t) - \gamma I_k(t) \\
&= b - (b + \mu + \gamma) N_k(t) + \gamma S_k(t) + \gamma R_k(t),
\end{aligned}$$

$$b - (b + \mu + \gamma) N_k(t) \le \frac{dN_k(t)}{dt} \le b - (b + \mu) N_k(t), \tag{2.2.2}$$

$$\limsup_{t \to \infty} (S_k(t) + I_k(t) + R_k(t)) = \limsup_{t \to \infty} N_k(t) \le b/(b + \mu). \tag{2.2.3}$$

Because of the initial value $N_k(0) \ge 0$, when $t \to \infty$, $N_k$ cannot increase to infinity, so let $\frac{dN_k(t)}{dt} \le 0$, then $b/(b + \mu + \gamma) \le \liminf_{t \to \infty} N_k(t) \le \limsup_{t \to \infty} N_k(t) \le b/(b + \mu)$, so, all feasible solutions are bounded in the domain $\Omega$. This completes the proof. □

**Lemma 3.** Define $\rho = \frac{b\beta}{(b+\mu)(\mu+\gamma+\sigma)} \frac{\langle k^2 \rangle}{\langle k \rangle}$, system (2.1.1) has a constant undisturbed equilibrium solution $E_0\left(b/(b+\mu), 0, 0\right)$ and when $\rho > 1$, it has a unique endemic equilibrium $E_+\left(S_k^\infty, I_k^\infty, R_k^\infty\right)$, where

$$
\begin{cases}
S_k^\infty = \frac{b(\mu+\delta)(\mu+\gamma+\sigma)}{(b+\mu)(\mu+\delta)(\mu+\gamma+\sigma)+\left[(\mu+\gamma+\sigma+b)(\mu+\delta)+(b-\delta)\sigma\right]\beta k\theta^\infty}, \\
I_k^\infty = \frac{b\beta k\theta^\infty(\mu+\delta)}{(b+\mu)(\mu+\delta)(\mu+\gamma+\sigma)+\left[(\mu+\gamma+\sigma+b)(\mu+\delta)+(b-\delta)\sigma\right]\beta k\theta^\infty}, \\
R_k^\infty = \frac{\sigma b\beta k\theta^\infty}{(b+\mu)(\mu+\delta)(\mu+\gamma+\sigma)+\left[(\mu+\gamma+\sigma+b)(\mu+\delta)+(b-\delta)\sigma\right]\beta k\theta^\infty}.
\end{cases}
$$

**Proof.** In the steady state, let the right side of system (2.1.1) be equal to zero, and the equilibrium point $E_+$ of system (2.1.1) satisfy:

$$
\begin{cases}
b\left(1 - S_k^\infty - I_k^\infty - R_k^\infty\right) - \beta k\theta^\infty S_k^\infty - \mu S_k^\infty + \delta R_k^\infty = 0, \\
\beta k\theta^\infty S_k^\infty - (\mu+\gamma)I_k^\infty - \sigma I_k^\infty = 0, \\
\sigma I_k^\infty - \mu R_k^\infty - \delta R_k^\infty = 0,
\end{cases}
\tag{2.2.4}
$$

$$
\theta^\infty = \frac{1}{\langle k \rangle} \sum_k kP(k)I_k^\infty.
\tag{2.2.5}
$$

First, we calculate the performance of the system (2.1.1), bring $I_k^\infty = 0, k = 1, 2, ..., k_{max}$ into (2.2.4), we can get the constant undisturbed equilibrium solution $E_0\left(b/(b+\mu), 0, 0\right)$. Suppose system (2.1.1) has a unique endemic equilibrium $E_+$, from (2.2.4), we obtain that:

$$
\begin{cases}
S_k^\infty = \frac{\mu+\gamma+\sigma}{\beta k\theta^\infty} I_k^\infty, \\
R_k^\infty = \frac{\sigma}{\mu+\delta} I_k^\infty.
\end{cases}
\tag{2.2.6}
$$

Inserting (2.2.6) into (2.2.4), we can get

$$
b\left(1 - \frac{\mu+\gamma+\sigma}{\beta k\theta^\infty} I_k^\infty - I_k^\infty - \frac{\sigma}{\mu+\delta} I_k^\infty\right) - (\mu+\gamma+\sigma)I_k^\infty - \mu\frac{\mu+\gamma+\sigma}{\beta k\theta^\infty} I_k^\infty + \delta\frac{\sigma}{\mu+\delta} I_k^\infty = 0.
\tag{2.2.7}
$$

By combining (2.2.6) with (2.2.7), we have

$$
\begin{cases}
S_k^\infty = \frac{b(\mu+\delta)(\mu+\gamma+\sigma)}{(b+\mu)(\mu+\delta)(\mu+\gamma+\sigma)+\left[(\mu+\gamma+\sigma+b)(\mu+\delta)+(b-\delta)\sigma\right]\beta k\theta^\infty}, \\
I_k^\infty = \frac{b\beta k\theta^\infty(\mu+\delta)}{(b+\mu)(\mu+\delta)(\mu+\gamma+\sigma)+\left[(\mu+\gamma+\sigma+b)(\mu+\delta)+(b-\delta)\sigma\right]\beta k\theta^\infty}, \\
R_k^\infty = \frac{b\sigma\beta k\theta^\infty}{(b+\mu)(\mu+\delta)(\mu+\gamma+\sigma)+\left[(\mu+\gamma+\sigma+b)(\mu+\delta)+(b-\delta)\sigma\right]\beta k\theta^\infty}.
\end{cases}
\tag{2.2.8}
$$

Inserting $I_k^\infty$ into (2.2.5), we can obtain

$$
\theta^\infty = \frac{1}{\langle k \rangle} \sum_k kP(k)I_k^\infty = \frac{\sum_k k^2 P(k)}{\langle k \rangle} \frac{b\beta\theta^\infty(\mu+\delta)}{(b+\mu)(\mu+\delta)(\mu+\gamma+\sigma)+\left((\mu+\gamma+\sigma+b)(\mu+\delta)+(b-\delta)\sigma\right)\beta k\theta^\infty} \triangleq f(\theta^\infty),
\tag{2.2.9}
$$

obviously, $\theta^\infty = 0$ is a solution of (2.2.9), i.e., $f(\theta^\infty) = 0$. Note that

$$f(1) = \frac{\sum_k k^2 P(k)}{\langle k \rangle} \frac{b\beta(\mu+\delta)}{(b+\mu)(\mu+\delta)(\mu+\gamma+\sigma)+((\mu+\gamma+\sigma+b)(\mu+\delta)+(b-\delta)\sigma)\beta k}$$

$$< \frac{\sum_k k^2 P(k)}{\langle k \rangle} \frac{b\beta(\mu+\delta)}{((\mu+\gamma+\sigma+b)(\mu+\delta)+(b-\delta)\sigma)\beta k}$$

$$< \frac{\sum_k k P(k)}{\langle k \rangle} \frac{b\beta k(\mu+\delta)}{b(\mu+\delta)\beta k}$$

$$= \frac{\sum_k k P(k)}{\langle k \rangle} = 1,$$

$$f'(\theta^\infty) = \frac{\sum_k k^2 P(k)}{\langle k \rangle} \frac{b\beta(b+\mu)(\mu+\delta)^2(\mu+\gamma+\sigma)}{\left((b+\mu)(\mu+\delta)(\mu+\gamma+\sigma)+((\mu+\gamma+\sigma+b)(\mu+\delta)+(b-\delta)\sigma)\beta k\theta^\infty\right)^2} > 0,$$

$$\quad (2.2.10)$$

$$f''(\theta^\infty) = -\frac{\sum_k k^2 P(k)}{\langle k \rangle} \frac{-2b\beta^2(b+\mu)(\mu+\delta)^2(\mu+\gamma+\sigma)\left((\mu+\gamma+\sigma+b)(\mu+\delta)+(b-\delta)\sigma\right)}{\left((b+\mu)(\mu+\delta)(\mu+\gamma+\sigma)+((\mu+\gamma+\sigma+b)(\mu+\delta)+(b-\delta)\sigma)\beta k\theta^\infty\right)^3} < 0.$$

Therefore, (2.2.9) has a unique positive solution if and only if

$$\frac{df(\theta^\infty)}{d\theta^\infty}\bigg|_{\theta^\infty = 0} = \frac{\sum_k k^2 P(k)}{\langle k \rangle} \frac{b\beta(b+\mu)(\mu+\delta)^2(\mu+\gamma+\sigma)}{\left((b+\mu)(\mu+\delta)(\mu+\gamma+\sigma)\right)^2} = \frac{b\beta}{(b+\mu)(\mu+\gamma+\sigma)} \frac{\langle k^2 \rangle}{\langle k \rangle} > 1,$$

$$\quad (2.2.11)$$

where $\langle k^2 \rangle = \sum_k k^2 P(k)$. Let

$$\rho = \frac{b\beta}{(b+\mu)(\mu+\gamma+\sigma)} \frac{\langle k^2 \rangle}{\langle k \rangle},$$

$$\quad (2.2.12)$$

if $\rho > 1$, we get that (2.2.9) has a nontrivial solution.

Then we can get the threshold of disturbance propagation

$$\beta_c = \frac{(b+\mu)(\mu+\gamma+\sigma)}{b} \frac{\langle k \rangle}{\langle k^2 \rangle},$$

$$\quad (2.2.13)$$

Substitute the nontrivial solution of (2.2.9) into (2.2.8), we can calculate $I_k^\infty$. By (2.2.6) and (2.2.7), we can also get $0 < S_k^\infty < 1$, $0 < I_k^\infty < 1$, $0 < R_k^\infty < 1$. So, the equilibrium $E_+ \left(S_k^\infty, I_k^\infty, R_k^\infty\right)$ is well defined. Hence, when $\rho > 1$, one and only one endemic equilibrium of system (2.1.1) exists. This completes the proof. □

**Remark**

(1) From Lemma 3 we can obtain that $\rho$ determines the existence of endemic equilibrium, and $\rho$ is called the basic reproductive number. To clarify its epidemiological meaning, the expression for $\rho$ (Equation 2.2.12) can be understood as the product of four components: (i) $\beta$, the transmission probability per contact; (ii) $S^0 = b/(b + \mu)$, the steady-state density of susceptible species in a disease-free environment; (iii) $1/(\mu + \gamma + \sigma)$, the average duration a species remains

in the disturbed (infectious) state before recovering or being removed; and (iv) $\langle k^2 \rangle / \langle k \rangle$, the effective connectivity of the heterogeneous network (excess degree). Thus, $\rho$ represents the average number of secondary disturbances generated by one disturbed species introduced into a fully undistributed ecosystem.

(2) From (2.2.12), we have that $\rho$ is related to $\beta$, $\sigma$ and the topological structure of the network. The increase of $\beta$ means that the stability of resistance decreases, so the more easily the disturbance propagates, the greater the value of $\rho$. The increase of $\sigma$ means that the stability of restoring force increases, so the disturbance is not easy to propagate and the value of $\rho$ is smaller.

(3) Namely, the disturbance incidence will die out if $\beta < \beta_c$, and it will break out if $\beta > \beta_c$. The threshold of disturbance propagation is related to the structure of ecological network, the rate of recovery and the proportion of emigrated species. For a given ecosystem, $\langle k^2 \rangle / \langle k \rangle$ is a definite real number. When $\sigma$ increases, the disturbance propagation threshold $\beta_c$ will rises but $I_k^\infty$ will decreases. That is to say, the easier the species recover from the disturbed state, the more difficult the disturbance propagation is. It needs a large enough propagation probability to maintain the disturbance propagation. At the same time, the increase of the proportion of species recovery indicates that the disturbance can be well controlled, and the proportion of disturbed species in the system will eventually decline. When $\mu$ and $\gamma$ increase, the disturbance propagation threshold $\beta_c$ will also rises but $I_k^\infty$ will decreases. In other words, the greater the proportion of species moving out of the ecosystem, the greater the impact of this disturbance on species, and the higher the proportion of species in the final disturbed state.

## 2.3 Global stability of equilibria

In this section, the stability of disease-free equilibrium $E_0 \left( b/ \left( b + \mu \right), 0, 0 \right)$ and endemic equilibrium $E_+ \left( S_k^\infty, I_k^\infty, R_k^\infty \right)$ will be analyzed. Firstly, we analyze the local asymptotic stability of the disease-free equilibrium, and then analyze its global stability. We will get that when $\rho < 1$, the disease-free equilibrium is globally stable, when $\rho > 1$, it is unstable.

**Theorem 1**. The disease-free equilibrium $E_0 \left( b/ \left( b + \mu \right), 0, 0 \right)$ of the system (2.1.1) is locally asymptotically stable if $\rho < 1$ and it is unstable if $\rho > 1$.

**Proof.** Let $S_k \left( t \right) = N_k \left( t \right) - I_k \left( t \right) - R_k \left( t \right)$, where $N_k \left( t \right) = b/ \left( b + \mu \right)$, we transform the system (2.1.1) as follows:

$$\begin{cases} \frac{dI_k(t)}{dt} = \frac{\beta k}{\langle k \rangle} \left( b/ \left( b + \mu \right) - I_k \left( t \right) - R_k \left( t \right) \right) \sum_k kP \left( k \right) I_k \left( t \right) - \left( \mu + \gamma + \sigma \right) I_k \left( t \right), \\ \frac{dR_k(t)}{dt} = \sigma I_k \left( t \right) - \mu R_k \left( t \right) - \delta R_k \left( t \right). \end{cases}$$

$$(2.3.1)$$

The Jacobian matrix of system (2.3.1) at $\left\{ \left( 0, 0 \right) \right\}$ is a $2k_{\max} \times 2k_{\max}$ matrix as follows:

$$J = \begin{bmatrix} A_1 & B_{12} & B_{13} & \cdots & B_{1k_{\max}} \\ B_{21} & A_2 & B_{23} & \cdots & B_{2k_{\max}} \\ \vdots & & \ddots & & \vdots \\ B_{k_{\max}1} & B_{k_{\max}2} & B_{k_{\max}3} & \cdots & A_{k_{\max}} \end{bmatrix},$$

where $A_j = \begin{bmatrix} \frac{\beta j^2}{\langle k \rangle} \frac{b}{b+\mu} P \left( j \right) - \left( \mu + \gamma + \sigma \right) & 0 \\ \sigma & - \left( \mu + \delta \right) \end{bmatrix}$, $B_{ij} = \begin{bmatrix} \frac{\beta ij}{\langle k \rangle} \frac{b}{b+\mu} P \left( j \right) & 0 \\ 0 & 0 \end{bmatrix}$.

Then we can get the characteristic polynomial of matrix $J$ by mathematical induction

$$\left( z + \left( \mu + \delta \right) \right)^{k_{\max}} \left( z + \left( \mu + \gamma + \sigma \right) \right)^{k_{\max}-1} \left( \left( z + \left( \mu + \gamma + \sigma \right) \right) - \beta \frac{b}{b + \mu} \frac{P \left( 1 \right) + 2^2 P \left( 2 \right) + \cdots + k_{\max}^2 P \left( k_{\max} \right)}{\langle k \rangle} \right),$$

$$(2.3.2)$$

this equation has a negative root $-(\mu + \delta)$ with multiplicity $k_{max}$ and a negative root $-(\mu + \gamma + \sigma)$ with multiplicity $k_{max} - 1$.

Note that:

$$\langle k^2 \rangle = \sum_k k^2 P(k) = \frac{P(1) + 2^2 P(2) + \cdots + k_{max}^2 P(k_{max})}{\langle k \rangle}. \tag{2.3.3}$$

So, the characteristic polynomial is equal to 0 if and only if

$$(z + (\mu + \gamma + \sigma)) - \beta \frac{b}{b + \mu} \frac{P(1) + 2^2 P(2) + \cdots + k_{max}^2 P(k_{max})}{\langle k \rangle} = 0. \tag{2.3.4}$$

If $\rho > 1$, $\beta \frac{b}{b+\mu} \frac{P(1) + 2^2 P(2) + \cdots + k_{max}^2 P(k_{max})}{\langle k \rangle} > \mu + \gamma + \sigma$, so $z > 0$ holds, and when $\rho < 1$, we can get $z < 0$. Hence $E_0$ is locally asymptotically stable if $\rho < 1$ and unstable if $\rho > 1$. This completes the proof. $\square$

**Lemma 4 [19].** If $a > 0$, $b > 0$ and $\frac{dx_k(t)}{dt} \geq b - ax$, when $t \geq 0$ and $x(0) \geq 0$, we have $\liminf_{t \to \infty} x(t) \geq \frac{b}{a}$. If $a > 0$, $b > 0$ and $\frac{dx_k(t)}{dt} \leq b - ax$, when $t \geq 0$ and $x(0) \geq 0$, we have $\limsup_{t \to \infty} x(t) \leq \frac{b}{a}$.

**Theorem 2.** The disease-free equilibrium $E_0$ of the system (2.1.1) is globally asymptotically stable when $\rho < 1$.

**Proof.** We rewrite the system (2.1.1) as

$$\begin{cases} \frac{dS_k(t)}{dt} = b\frac{\mu}{b+\mu} - \beta k\theta(t) S_k(t) - (\mu + \delta) S_k(t) + \delta \left( \frac{b}{b+\mu} - I_k(t) \right), \\ \frac{dI_k(t)}{dt} = \beta k\theta(t) S_k(t) - (\mu + \gamma) I_k(t) - \sigma I_k(t). \end{cases} \tag{2.3.5}$$

Let $\{S_k(t), I_k(t), R_k(t)\}_1^{k_{max}}$ be a nonnegative solution of system (2.3.5), from $\lim_{t \to \infty} I_k(t) = 0$ and the first equation of system (2.3.5), we have

$$\frac{dS_k(t)}{dt} \leq \frac{b\mu}{b+\mu} + \frac{b\delta}{b+\mu} - (\mu + \delta) S_k(t) = (\mu + \delta) \frac{b}{b+\mu} - (\mu + \delta) S_k(t).$$

By Lemma 4, we have

$$\limsup_{t \to \infty} S_k(t) \leq \frac{b}{b+\mu} =: S_k^0. \tag{2.3.6}$$

Thus, for arbitrarily enough small $\varepsilon_1 > 0$, there exists $t_1 > 0$ such that $S_k(t) \leq S_k^0 + \varepsilon_1$ for $t > t_1$. If $t > t_1$, from the second equation of system (2.3.5) we have

$$\frac{dI_k(t)}{dt} \leq \beta k\theta(t) \left( S_k^0 + \varepsilon_1 \right) - (\mu + \gamma + \sigma) I_k(t).$$

Now we consider the following comparison system with initial condition $U_k(0) = I_k(0) \geq 0$

$$\frac{dU_k(t)}{dt} = \beta k\theta_1(t) \left( S_k^0 + \varepsilon_1 \right) - (\mu + \gamma + \sigma) U_k(t), \text{ where } \theta_1(t) = \sum_k \frac{kP(k) U_k(t)}{\langle k \rangle}. \tag{2.3.7}$$

Now we prove the positive solutions of (2.3.7) tend to zero as $t \to \infty$. Construct the Lyapunov function as follows

$$V(t) = \sum_k \alpha_k U(t) \text{ where } \alpha_k = \frac{kP(k)}{(\mu+\gamma+\sigma)\langle k \rangle}. \quad (2.3.8)$$

Then we have

$$\frac{dV}{dt} \leq \sum_k \alpha_k \left( \beta k \theta_1(t) \left( S_k^0 + \varepsilon_1 \right) - (\mu + \gamma + \sigma)U_k(t) \right)$$

$$= \sum_k \left( \frac{kP(k)}{(\mu + \gamma + \sigma)\langle k \rangle} \beta k \theta_1(t) \left( S_k^0 + \varepsilon_1 \right) - \frac{kP(k)}{\langle k \rangle} U_k(t) \right)$$

$$= \theta_1(t) \left( \rho + \frac{\beta \langle k^2 \rangle}{(\mu + \gamma + \sigma)\langle k \rangle} \varepsilon_1 - 1 \right).$$

If $\rho < 1$, there exists an $\varepsilon_1 > 0$ small enough such that $\rho + \frac{\beta \langle k^2 \rangle}{(\mu+\gamma+\sigma)\langle k \rangle}\varepsilon_1 < 1$, then we have $\rho < 1$, $\frac{dV}{dt} \leq 0$ for $U_k(t) \geq 0$, and that $\frac{dV}{dt} = 0$ only if $U_k(t) = 0$. Thus, we can get the positive solutions of (2.3.7) tend to zero as $t \to \infty$, that is $\lim_{t \to \infty} U_k(t) = 0$. By the comparison theorem, we have $0 \leq I_k(t) \leq U_k(t)$ for all $t > 0$. Therefore $\lim_{t \to \infty} I_k(t) = 0$ for $k = 1, \cdots, k_{max}$.

Now we prove $\lim_{t \to \infty} S_k(t) = S_k^0$. Since $\lim_{t \to \infty} I_k(t) = 0$, there exists an $\varepsilon_2 > 0$ arbitrarily small enough and a $t_2 > 0$ such that $0 \leq I_k(t) \leq \varepsilon_2$ for $t > t_2$. From the first equation of system (2.3.5), we have

$$\frac{dS_k(t)}{dt} \geq \frac{b\mu}{b+\mu} + \frac{b\delta}{b+\mu} - \delta\varepsilon_2 - (\mu + \delta + A\varepsilon_2)S_k(t), \text{ where } A = \frac{\beta k}{\langle k \rangle}\sum_k kP(k). \quad (2.3.9)$$

By Lemma 4, we have $\liminf_{t \to \infty} S_k(t) \geq \frac{b(\mu+\delta)-(b+\mu)\delta\varepsilon_2}{(b+\mu)(\mu+\delta+A\varepsilon_2)}$, setting $\varepsilon_2 \to 0$, it follows that

$$\liminf_{t \to \infty} S_k(t) \geq \frac{b}{(b+\mu)} = S_k^0, \quad (2.3.10)$$

From (2.3.6) and (2.3.10), obviously $\lim_{t \to \infty} S_k(t) = S_k^0 = \frac{b}{(b+\mu)}$.

Finally, from $R_k(t) = \frac{b}{b+\mu} - S_k(t) - I_k(t)$, we have $\lim_{t \to \infty} R_k(t) = R_k^0 = 0$. This prove that the disease-free equilibrium $E_0$ of system (2.1.1) is globally attractive when $\rho < 1$. Because of the local stability and global attractivity of $E_0$, we obtain that $E_0$ is globally stable when $\rho < 1$ by the LaSalle's invariant principle [20]. This completes the proof. □

**Theorem 3.** When $\rho > 1$, the disturbance is persistent in the ecological networks, i.e., there exists $\zeta > 0$, such that:

$$\liminf_{t \to \infty} I(t) = \liminf_{t \to \infty} \sum_k P(k) I_k(t) > \zeta.$$

**Proof.** Now we investigate the persistence conditions of perturbations on the networks by Theorem 4.6 [21]. Define:

$$X = \left\{ (S_1, I_1, R_1, \cdots, S_{k_{max}}, I_{k_{max}}, R_{k_{max}}) : S_k, I_k, R_k \geq 0 \quad and \quad S_k + I_k + R_k = \frac{b}{b+\mu}, \quad k = 1, \cdots, k_{max} \right\},$$

$$X_0 = \left\{ (S_1, I_1, R_1, \cdots, S_{k_{max}}, I_{k_{max}}, R_{k_{max}}) \in X : \sum_k P(k)I_k > 0 \right\},$$

$$\partial X_0 = X \backslash X_0.$$

The next step is to prove that system (2.1.1) is uniformly persistent in $X_0$ and $\partial X_0$.

Obviously, $X$ is positively invariant in system (2.1.1). If $S_k(0) \geq 0$, $\sum_k P(k)I_k(0) > 0$ and $R_k(0) \geq 0$ for $k = 1, \cdots, k_{\max}$, then we have that $S_k(t) \geq 0$, $\sum_k P(k)I_k(t) > 0$, and $R_k(t) \geq 0$ for all $t > 0$. Through calculation, we can get $\left(\sum_k P(k)I_k(t)\right)' \geq -(\mu + \gamma + \sigma)\sum_k P(k)I_k(t)$ and $\sum_k P(k)I_k(0) > 0$, so $\sum_k P(k)I_k(t) \geq \sum_k P(k)I_k(0)\, e^{-(\sigma+\gamma+\mu)} > 0$. Therefore, $X_0$ is also positively invariant. In addition, all solutions of system (2.1.1) in $X$ are contained in a compact set B and exist permanently. Next, we will prove the compactness condition of set B. Denote:

$$M_\partial = \left\{ \left(S_1(0), I_1(0), R_1(0), \cdots, S_{k_{\max}}(0), I_{k_{\max}}(0), R_{k_{\max}}(0)\right) : \right.$$
$$\left. \left(S_1(t), I_1(t), R_1(t), \cdots, S_{k_{\max}}(t), I_{k_{\max}}(t), R_{k_{\max}}(t)\right) \in \partial X_0, t \geq 0 \right\},$$

$$\Omega = \bigcup \left\{ \omega\left(S_1(0), I_1(0), R_1(0), \cdots, S_{k_{\max}}(0), I_{k_{\max}}(0), R_{k_{\max}}(0)\right) : \right.$$
$$\left. \left(S_1(0), I_1(0), R_1(0), \cdots, S_{k_{\max}}(0), I_{k_{\max}}(0), R_{k_{\max}}(0)\right) \in X \right\}.$$

where $\omega\left(S_1(0), I_1(0), R_1(0), \cdots, S_{k_{\max}}(0), I_{k_{\max}}(0), R_{k_{\max}}(0)\right)$ is the limit set of solutions of system (2.1.1) starting from $\left(S_1(0), I_1(0), R_1(0), \cdots, S_{k_{\max}}(0), I_{k_{\max}}(0), R_{k_{\max}}(0)\right)$. The restriction condition of system (2.1.1) on $M_\partial$ is given:

$$\begin{cases} \frac{dS_k(t)}{dt} = b - (b + \mu)S_k(t) - bI_k(t) + (\delta - b)R_k(t), \\ \frac{dI_k(t)}{dt} = -(\mu + \gamma + \sigma)I_k(t), \\ \frac{dR_k(t)}{dt} = \sigma I_k(t) - (\mu + \delta)R_k(t). \end{cases} \tag{2.3.11}$$

Obviously, system (2.3.11) has only one equilibrium $E_0$ in $X$, then we have the system (2.1.1) in $M_\partial$ has the unique equilibrium $E_0$. Next, we will prove that equilibrium $E_0$ is locally asymptotically stable. In other words, for system (2.3.11), $E_0$ is globally asymptotically stable. Thus $\Omega = \{E_0\}$, it is a single point set. Finally, to prove that $E_0$ is locally asymptotically stable, we only need to prove it is a weak repulsion for $X_0$, i.e.,

$$\limsup_{t \to \infty} \ dist\left(S_1(t), I_1(t), R_1(t), \cdots, S_{k_{\max}}(t), I_{k_{\max}}(t), R_{k_{\max}}(t), E_0\right) > 0,$$

where $\left(S_1(t), I_1(t), R_1(t), \cdots, S_{k_{\max}}(t), I_{k_{\max}}(t), R_{k_{\max}}(t)\right)$ is an arbitrarily solution with initial value in $X_0$. Next, we just need to prove the intersection between $E_0$ and the stable manifold of $E_0$ is an empty set, i.e., $W^s(E_0) \bigcap E_0 = \varnothing$. We adopt the idea of proof to the contrary, assume that the above formula does not hold, then there exists a solution $\left(S_1(t), I_1(t), R_1(t), \cdots, S_{k_{\max}}(t), I_{k_{\max}}(t), R_{k_{\max}}(t)\right)$ in $X_0$, such that:

$$S_k(t) \to b/(b + \mu), I_k(t) \to 0, R_k(t) \to 0 \quad \text{as } t \to \infty.$$

Since $\rho = \frac{b\beta}{(b+\mu)(\mu+\gamma+\sigma)}\frac{\langle k^2 \rangle}{\langle k \rangle} > 1$, convert the above formula into $\sum_k \frac{\beta k^2 P(k)}{\langle k \rangle}\frac{b}{b + \mu} > \mu + \gamma + \sigma$, we can take $\varsigma > 0$ satisfying the following formula:

$$\langle \frac{\beta k^2 P(k)}{\langle k \rangle}\left(\frac{b}{b + \mu} - \varsigma\right)\rangle > (\mu + \gamma + \sigma) \tag{2.3.12}$$

For $\varsigma > 0$, there exists a $T > 0$ such that $b/(b + \mu) - \varsigma < S_k(t) < b/(b + \mu) + \varsigma, 0 < I_k(t) < \varsigma, 0 < R_k(t) < \varsigma$ for all $t \geq T$ and $k = 1, \cdots, k_{\max}$. Let $V(t) = \sum_k kP(k)I_k(t)$.

We can calculate the derivative of $V$ along the solution $(S_1(t), I_1(t), R_1(t), \cdots, S_{k_{\max}}(t), I_{k_{\max}}(t), R_{k_{\max}}(t))$:

$$V'(t) = \sum_k kP(k)\left[\beta k S_k(t)\frac{\sum_k kP(k)I_k(t)}{\langle k\rangle} - (\mu+\gamma+\sigma)I_k(t)\right]$$

$$\geq \sum_k P(k)\frac{\beta k^2}{\langle k\rangle}(b/(b+\mu)-\varsigma)\sum_k kP(k)I_k(t) - \sum_k kP(k)(\mu+\gamma+\sigma)I_k(t)$$

$$= \sum_k P(k)k\left[\langle\frac{\beta k^2 P(k)}{\langle k\rangle}\left(\frac{b}{b+\mu}-\varsigma\right)\rangle - (\mu+\gamma+\sigma)\right]I_k(t) \geq \alpha V(t) \geq 0.$$

(2.3.13)

Where $\alpha=\langle\frac{\beta k^2 P(k)}{\langle k\rangle}\left(\frac{b}{b+\mu}-\varsigma\right)\rangle - (\mu+\gamma+\sigma) > 0$.

Hence, we get $V(t) > V(0)e^{\alpha t}$, from $\theta(t) > 0$, we have $V(0) = \sum_k kP(k)I_k(0) > 0$, so $V(t)\to\infty$ as $t\to\infty$, which

contradicts to the boundedness of $V(t)$. This completes the proof. □

Now we prove the global asymptotical stability of the unique endemic equilibrium $E_+$ according to the LaSalle's invariant principle [20].

**Theorem 4**. The unique endemic equilibrium $E_+(S_k^\infty, I_k^\infty, R_k^\infty)$ of the system (2.1.1) is locally asymptotically stable when $\rho > 1$.

**Proof.** Let $\{(S_k, I_k, R_k)\}_{k=1}^{k_{\max}}$ be the positive solution of system (2.1.1), let $x_k(t) = S_k(t) - S_k^\infty$, $y_k(t) = I_k(t) - I_k^\infty$, $z_k(t) = R_k(t) - R_k^\infty$, we have

$$\begin{cases} \frac{dx_k}{dt} = -b(x_k+y_k+z_k) - \frac{\beta k x_k}{\langle k\rangle}\sum_k kP(k)y_k - \beta k\theta^\infty x_k - \frac{\beta k S_k^\infty}{\langle k\rangle}\sum_k kP(k)y_k - \mu x_k + \delta z_k, \\ \frac{dy_k}{dt} = \frac{\beta k x_k}{\langle k\rangle}\sum_k kP(k)y_k + \beta k\theta^\infty x_k + \frac{\beta k S_k^\infty}{\langle k\rangle}\sum_k kP(k)y_k - (\mu+\gamma+\sigma)y_k, \\ \frac{dz_k}{dt} = \sigma y_k - (\mu+\delta)z_k. \end{cases}$$

(2.3.14)

So, the local asymptotic stability of $E_+(S_k^\infty, I_k^\infty, R_k^\infty)$ equivalent to the local asymptotic stability of the zero solution of system (2.1.1). Now we prove that the zero solution of system (2.3.14) is locally asymptotically stable.

Let $u_k = (x_k, y_k, z_k)$, $u = (u_1, u_2, ..., u_{k_{\max}})$, and define function $V_i(u)$ $(i = 1, 2, ..., 4)$ as follows

$$V_1(u) = \frac{1}{2}\sum_k \alpha_k(x_k+y_k)^2, \quad V_2(u) = \frac{1}{2}\sum_k \beta_k y_k^2,$$

$$V_3(u) = \frac{1}{2}\sum_k \xi_k z_k^2, \quad V_4(u) = \frac{1}{2}\sum_k \gamma_k(x_k+y_k+z_k)^2,$$

Where $\alpha_k, \beta_k, \xi_k, \gamma_k$ are pending normal numbers.

From system (2.3.14), we have

$$\frac{dV_1}{dt} = \sum_k \alpha_k(x_k+y_k)\left[-(b+\mu)x_k - (b+\mu+\gamma+\sigma)y_k + (\delta-b)z_k\right]$$

$$= \sum_k \alpha_k\left[-(b+\mu)x_k^2 - F_1 x_k y_k - F_2 y_k^2 + (\delta-b)z_k(x_k+y_k)\right],$$

(2.3.15)

where $F_1 = 2b + 2\mu + \gamma + \sigma$, $F_2 = b + \mu + \gamma + \sigma$.

$$\frac{dV_2}{dt} = \sum_k \beta_k y_k \left[ \frac{\beta k x_k}{\langle k \rangle} \sum_k kP(k) y_k + \beta k \theta^\infty x_k + \frac{\beta k S_k^\infty}{\langle k \rangle} \sum_k kP(k) y_k - (\mu + \gamma + \sigma) y_k \right]$$

$$= H(u) + \theta^\infty \sum_k \beta k \beta_k x_k y_k + \frac{1}{\langle k \rangle} \sum_k \beta k \beta_k S_k^\infty y_k \sum_k kP(k) y_k - F_3 \sum_k \beta_k y_k,$$

(2.3.16)

where $H(u) = \frac{1}{\langle k \rangle} \sum_k \beta k \beta_k x_k y_k \sum_k kP(k) y_k$, $F_3 = \mu + \gamma + \sigma$.

$$\frac{dV_3}{dt} = \sum_k \xi_k z_k \left[ \sigma y_k - (\mu + \delta) z_k \right]$$

$$= \sum_k \xi_k \left[ \sigma y_k z_k - (\mu + \delta) z_k^2 \right],$$

(2.3.17)

$$\frac{dV_4}{dt} = \frac{1}{2} \sum_k \gamma_k (x_k + y_k + z_k) \left[ -(b + \mu)(x_k + y_k + z_k) - \gamma y_k \right]$$

$$\leq \sum_k \left( -\gamma x_k y_k - \gamma y_k^2 - \gamma y_k z_k \right).$$

(2.3.18)

Young inequality $ab \leq \frac{a^2}{2\varepsilon} + \frac{\varepsilon b^2}{2} \ (\varepsilon > 0)$, from above inequality we have

$$\frac{dV_1}{dt} \leq \sum_k \alpha_k \left[ -\frac{1}{2}(b + \mu) x_k^2 - F_1 x_k y_k - \frac{1}{2} F_2 y_k^2 + F_4 z_k^2 \right],$$

(2.3.19)

where $F_4 = \frac{F_1 (\delta - b)^2}{2 F_2 (b + \mu)}$.

Cauchy inequality $\left( \sum_k a_k b_k \right)^2 \leq \sum_k a_k^2 \sum_k b_k^2$, we have

$$\left| \sum_k \beta k \beta_k S_k^\infty y_k \right| \leq \sqrt{\sum_k \beta_k (\beta k S_k^\infty)^2} \sqrt{\sum_k \beta_k y_k^2},$$

$$\left| \sum_k kP(k) y_k \right| \leq \sqrt{\sum_k \frac{k^2 P^2(k)}{\beta_k}} \sqrt{\sum_k \beta_k y_k^2}.$$

Let $\beta_k = \frac{P(k)}{\beta S_k^\infty}$, from $\frac{\beta}{\langle k \rangle} \sum_k k^2 P(k) S_k^\infty = \mu + \gamma + \sigma = F_3$, we have

$$\frac{1}{\langle k \rangle} \sum_k \beta k \beta_k S_k^\infty y_k \sum_k kP(k) y_k \leq \sum_k k^2 P(k) S_k^\infty \sum_k \beta_k y_k^2 = F_3 \sum_k \beta_k y_k^2.$$

(2.3.20)

Combining (2.3.16) and (2.3.20), we have

$$\frac{dV_2}{dt} \leq H(u) + \theta^\infty \sum_k \beta k \beta_k x_k y_k, \tag{2.3.21}$$

let $\alpha_k = \frac{\beta k \beta_k \theta^\infty}{F_1}$, we have

$$\frac{d(V_1 + V_2)}{dt} \leq \sum_k \alpha_k \left[ -\frac{1}{2}(b+\mu)x_k^2 - \frac{1}{2}F_2 y_k^2 + F_4 z_k^2 \right] + H(u), \tag{2.3.22}$$

let $\xi_k = \frac{\gamma \gamma_k}{\sigma}$ and $\gamma_k = \frac{\beta k \beta_k \theta^\infty}{\gamma}$, we have

$$\frac{d(V_2 + V_3 + V_4)}{dt} \leq -\sum_k \gamma_k \left[ \gamma y_k^2 + \frac{\gamma(\mu+\delta)}{\sigma} z_k^2 \right] + H(u), \tag{2.3.23}$$

let $V(u) = A(V_2 + V_3 + V_4) + (V_1 + V_2)$, where $A = \frac{2F_4 \sigma}{F_1(\mu+\delta)}$, and $\gamma_k = \frac{F_1 \alpha_k}{\gamma}$, from (2.3.22) and (2.3.23), we have

$$\frac{dV}{dt} \leq \sum_k \alpha_k \left[ -\frac{1}{2}(b+\mu)x_k^2 - \frac{1}{2}F_2 y_k^2 - F_4 z_k^2 \right] + H(u)$$
$$= -\sum_k \left( A_k x_k^2 + B_k y_k^2 + C_k z_k^2 \right) + H(u),$$

where $A_k = \frac{\alpha_k}{2}(b+\mu)$, $B_k = \frac{\alpha_k}{2}F_2$, $C_k = F_4 \alpha_k$. Let $\theta = \min_{1 \leq k \leq k_{max}} \{A_k, B_k, C_k\}$, then $\theta > 0$, we have

$$\frac{dV}{dt} \leq -\theta |u(t)|^2 + o\left( |u(t)|^2 \right),$$

where $u(t) = \sqrt{\sum_k \left( x_k^2 + y_k^2 + z_k^2 \right)}$, $o\left( |u(t)|^2 \right)$ is the infinitesimal of higher order of $|u(t)|^2$. Thus $V'$ is negative definite in a neighborhood of $u = 0$. Then we can get that the zero solution of system (2.3.14) is locally asymptotically stable and $E_+ \left( S_k^\infty, I_k^\infty, R_k^\infty \right)$ is locally asymptotically stable. This completes the proof. □

**Theorem 5**. The unique endemic equilibrium $E_+ \left( S_k^\infty, I_k^\infty, R_k^\infty \right)$ of the system (2.1.1) is global asymptotically stable when $\rho > 1$.

**Proof.** Before detailing the mathematical derivation, we outline the intuition behind the proof, which utilizes the Monotone Iteration Technique. We construct two sequences of limiting values to bound the system's trajectory: an upper sequence $(W_{k,m}, X_{k,m}, \ldots)$ that decreases monotonically and a lower sequence $(w_{k,m}, x_{k,m}, \ldots)$ that increases monotonically. By analyzing the limit sets of the system, we show that as the iteration count $m \to \infty$, both sequences converge to the unique endemic equilibrium $E_+$. Consequently, the solution $\left( S_k(t), I_k(t), R_k(t) \right)$ is 'squeezed' between these converging bounds, forcing it to globally stabilize at $E_+$.

First, we prove that the unique endemic equilibrium $E_+ \left( S_k^\infty, I_k^\infty, R_k^\infty \right)$ is globally attractive, that is

$\lim_{t \to \infty} S_k(t) = S_k^\infty$, $\lim_{t \to \infty} I_k(t) = I_k^\infty$, $\lim_{t \to \infty} R_k(t) = R_k^\infty$. From Theorem 3, there exists a sufficiently small constant $\zeta > 0$ and a sufficiently large constant $T_0 > 0$, such that

$$\xi \leq \theta(t) \leq 1, \quad \forall t > T_0. \tag{2.3.24}$$

From Lemma 2, we can get that for arbitrarily constant $0 < \zeta_1 < \mu/2\,(b + \mu)$, there exists a constant $T_1 > T_0$ such that

$$N_k(t) \le W_{k,1} - \zeta_1, \forall t > T_1,\tag{2.3.25}$$

where $W_{k,1} = \frac{b}{b+\mu} + 2\zeta_1 < 1$. From the first equation of system (2.1.1), we have

$$\frac{dS_k(t)}{dt} \le b\,(1 - S_k(t)) - \beta k \zeta S_k(t) - \mu S_k(t) + \delta\,(N_k(t) - S_k(t))$$
$$\le b + \delta W_{k,1} - (b + \beta k \zeta + \mu + \delta)\,S_k(t),$$

so, for arbitrarily constant $0 < \zeta_2 < \min\left(\frac{1}{3}, \zeta_1, \frac{W_{k,1}(b+\beta k\zeta+\mu)-b}{2(b+\beta k\zeta+\mu+\delta)}\right)$, there exists a constant $T_2 > T_1$ such that

$$S_k(t) \le X_{k,1} - \zeta_2, \forall t > T_2,$$

where $X_{k,1} = \frac{b+\delta W_{k,1}}{b+\beta k\zeta+\mu+\delta} + 2\zeta_2 < W_{k,1}$.

From the second equation of system (2.1.1), we have

$$\frac{dI_k(t)}{dt} \le \beta k\,(N_k(t) - I_k(t)) - (\mu + \gamma + \sigma)\,I_k(t)$$
$$\le \beta k W_{k,1} - (\beta k + \mu + \gamma + \sigma)\,I_k(t), \forall t > T_2,$$

so, for arbitrarily constant $0 < \zeta_3 < \min\left(\frac{1}{4}, \zeta_2, \frac{W_{k,1}(\mu+\gamma+\sigma)}{2(\beta k+\mu+\gamma+\sigma)}\right)$, there exists a constant $T_3 > T_2$ such that

$$I_k(t) \le Y_{k,1} - \zeta_3, \forall t > T_3,$$

where $Y_{k,1} = \frac{\beta k W_{k,1}}{\beta k+\mu+\gamma+\sigma} + 2\zeta_3 < W_{k,1}$.

From the third equation of system (2.1.1), we have

$$\frac{dR_k(t)}{dt} \le \sigma\,(N_k(t) - R_k(t)) - (\mu + \delta)\,R_k(t)$$
$$\le \sigma W_{k,1} - (\sigma + \mu + \delta)\,R_k(t), \forall t > T_3,$$

so, for arbitrarily constant $0 < \zeta_4 < \min\left(\frac{1}{5}, \zeta_3, \frac{W_{k,1}(\mu+\delta)}{2(\sigma+\mu+\delta)}\right)$, there exists a constant $T_4 > T_3$ such that

$$R_k(t) \le Z_{k,1} - \zeta_4, \forall t > T_4,$$

where $Z_{k,1} = \frac{\sigma W_{k,1}}{\sigma+\mu+\delta} + 2\zeta_4 < W_{k,1}$.

From Lemma 2, we can get that for arbitrarily constant $0 < \zeta_5 < \min\left(\frac{1}{6}, \zeta_4, \frac{b}{2(b+\mu+\gamma)}\right)$, there exists a constant $T_5 > T_4$ such that

$$N_k(t) \ge w_{k,1} + \zeta_5, \forall t > T_5,\tag{2.3.26}$$

where $w_{k,1} = \frac{b}{b+\mu+\gamma} - 2\zeta_5 > 0$. From the first equation of system (2.1.1), we have

$$\frac{dS_k(t)}{dt} \ge b\,(1 - W_{k,1}) - \beta k S_k(t) - \mu S_k(t),$$

so, for arbitrarily constant $0 < \zeta_6 < \min\left(\frac{1}{7}, \zeta_5, \frac{b(1-w_{k,1})}{2(\beta k+\mu)}\right)$, there exists a constant $T_6 > T_5$ such that

$$S_k(t) \geq x_{k,1} + \zeta_6, \forall t > T_6,$$

where $x_{k,1} = \frac{b(1-w_{k,1})}{\beta k+\mu} - 2\zeta_6 > 0$.

From the second equation of system (2.1.1), we have

$$\frac{dI_k(t)}{dt} \geq \beta k \zeta x_{k,1} - (\mu + \gamma + \sigma) I_k(t),$$

so, for arbitrarily constant $0 < \zeta_7 < \min\left(\frac{1}{8}, \zeta_6, \frac{\beta k \zeta x_{k,1}}{2(\mu+\gamma+\sigma)}\right)$, there exists a constant $T_7 > T_6$ such that

$$I_k(t) \geq y_{k,1} + \zeta_7, \forall t > T_7,$$

where $y_{k,1} = \frac{\beta k \zeta x_{k,1}}{\mu+\gamma+\sigma} - 2\zeta_7 > 0$.

From the third equation of system (2.1.1), we have

$$\frac{dR_k(t)}{dt} \geq \sigma y_{k,1} - (\mu + \delta) R_k(t), \forall t > T_7,$$

so, for arbitrarily constant $0 < \zeta_8 < \min\left(\frac{1}{9}, \zeta_7, \frac{\sigma y_{k,1}}{2(\mu+\delta)}\right)$, there exists a constant $T_8 > T_7$ such that

$$R_k(t) \geq z_{k,1} + \zeta_8, \forall t > T_8,$$

where $z_{k,1} = \frac{\sigma y_{k,1}}{\mu+\delta} - 2\zeta_8 > 0$.

From $\frac{dN_k(t)}{dt} \leq b - (b + \mu + \gamma) N_k(t) + \gamma X_{k,1} + \gamma Z_{k,1}, \forall t > T_8$, we have that for arbitrarily constant $0 < \zeta_9 < \min\left(\frac{1}{10}, \zeta_8\right)$, there exists a constant $T_9 > T_8$ such that

$$N_k(t) \leq W_{k,2}, \forall t > T_9,$$

where $W_{k,2} = \min\left(W_{k,1} - \zeta_1, \frac{b+\gamma X_{k,1}+\gamma Z_{k,1}}{b+\mu+\gamma} + \zeta_9\right)$. From the first equation of system (2.1.1), we have

$$\frac{dS_k(t)}{dt} \leq b(1-w_{k,1}) - \beta k \underline{\theta}_1 S_k(t) - \mu S_k(t) + \delta Z_{k,1}, \forall t > T_9,$$

where $\underline{\theta}_1 = \dfrac{\sum\limits_k kP(k)y_{k,1}}{\langle k \rangle}$, so, for arbitrarily constant $0 < \zeta_{10} < \min\left(\frac{1}{11}, \zeta_9\right)$, there exists a constant $T_{10} > T_9$ such that

$$S_k(t) \leq \min\left(X_{k,1} - \zeta_2, \frac{b(1-w_{k,1}) + \delta Z_{k,1}}{\beta k \underline{\theta}_1 + \mu} + \zeta_{10}\right) =: X_{k,2}, \forall t > T_{10}.$$

From the second equation of system (2.1.1), we have

$$\frac{dI_k(t)}{dt} \leq \beta k X_{k,2} \bar{\theta}_1 - (\mu + \gamma + \sigma) I_k(t), \forall t > T_{10},$$

where $\overline{\theta}_1 = \dfrac{\sum\limits_k kP(k)Y_{k,1}}{\langle k \rangle}$, so, for arbitrarily constant $0 < \zeta_{11} < \min\left(\frac{1}{12}, \zeta_{10}\right)$, there exists a constant $T_{11} > T_{10}$ such that

$I_k(t) \leq \min\left(Y_{k,1} - \zeta_3, \frac{\beta k X_{k,2}\overline{\theta}_1}{\mu+\gamma+\sigma} + \zeta_{11}\right) =: Y_{k,2}, \forall t > T_{11}$.

From the third equation of system (2.1.1), we have

$$\frac{dR_k(t)}{dt} \leq \sigma Y_{k,2} - (\mu + \delta) R_k(t)$$

so, for arbitrarily constant $0 < \zeta_{12} < \min\left(\frac{1}{13}, \zeta_{11}\right)$, there exists a constant $T_{12} > T_{11}$ such that

$$R_k(t) \leq \min\left(Z_{k,1} - \zeta_4, \frac{\sigma Y_{k,2}}{\mu+\sigma} + \zeta_{12}\right) =: Z_{k,2}, \forall t > T_{12}.$$

From $\frac{dN_k(t)}{dt} \geq b - (b + \mu + \gamma) N_k(t) + \gamma x_{k,1} + \gamma z_{k,1}, \forall t > T_{12}$,
we have that for arbitrarily constant $0 < \zeta_{13} < \min\left(\frac{1}{14}, \zeta_{12}, \frac{b+\gamma x_{k,1}+\gamma z_{k,1}}{2(b+\mu+\gamma)}\right)$, there exists a constant $T_{13} > T_{12}$ such that

$$N_k(t) \geq w_{k,2} + \zeta_{13}, \forall t > T_{13},$$

where $w_{k,2} = \max\left(w_{k,1} + \zeta_5, \frac{b+\gamma x_{k,1}+\gamma z_{k,1}}{b+\mu+\gamma} - 2\zeta_{13}\right)$. From the first equation of system (2.1.1), we have

$$\frac{dS_k(t)}{dt} \geq b(1 - W_{k,1}) - \beta k\overline{\theta}_1 S_k(t) - \mu S_k(t) + \delta z_{k,1}, \forall t > T_{13},$$

so, for arbitrarily constant $0 < \zeta_{14} < \min\left(\frac{1}{15}, \zeta_{13}, \frac{b(1-W_{k,1})+\delta z_{k,1}}{2(\beta k\overline{\theta}_1+\mu)}\right)$, there exists a constant $T_{14} > T_{13}$ such that

$$S_k(t) \geq x_{k,2} + \zeta_{14}, \forall t > T_{14},$$

where $x_{k,2} = \max\left(x_{k,1} + \zeta_6, \frac{b(1-W_{k,1})+\delta z_{k,1}}{\beta k\overline{\theta}_1+\mu} - 2\zeta_{14}\right)$.
From the second equation of system (2.1.1), we have

$$\frac{dI_k(t)}{dt} \geq \beta k X_{k,2}\underline{\theta}_1 - (\mu + \gamma + \sigma) I_k(t), \forall t > T_{14},$$

so, for arbitrarily constant $0 < \zeta_{15} < \min\left(\frac{1}{16}, \zeta_{14}, \frac{\beta k X_{k,2}\theta_1}{2(\mu+\gamma+\sigma)}\right)$, there exists a constant $T_{15} > T_{14}$ such that
$I_k(t) \geq y_{k,2} + \zeta_{15}, \forall t > T_{15}$,
where $y_{k,2} = \max\left(y_{k,1} + \zeta_7, \frac{\beta k X_{k,2}\theta_1}{\mu+\gamma+\sigma} - 2\zeta_{15}\right)$
From the third equation of system (2.1.1), we have

$$\frac{dR_k(t)}{dt} \geq \sigma y_{k,2} - (\mu + \delta) R_k(t), \forall t > T_{16},$$

so, for arbitrarily constant $0 < \zeta_{16} < \min\left(\frac{1}{17}, \zeta_{15}, \frac{\sigma y_{k,2}}{2(\mu+\delta)}\right)$, there exists a constant $T_{16} > T_{15}$ such that

$$R_k(t) \geq z_{k,2} + \zeta_{16}, \forall t > T_{16},$$

where $z_{k,2} = \max\left(z_{k,1} + \zeta_8, \frac{\sigma y_{k,2}}{\mu+\delta} - 2\zeta_{16}\right)$.

Repeating the above process, we have eight sequences as follows

$$\{W_{k,m}\}_{m=1}^{\infty}, \{X_{k,m}\}_{m=1}^{\infty}, \{Y_{k,m}\}_{m=1}^{\infty}, \{Z_{k,m}\}_{m=1}^{\infty},$$
$$\{w_{k,m}\}_{m=1}^{\infty}, \{x_{k,m}\}_{m=1}^{\infty}, \{y_{k,m}\}_{m=1}^{\infty}, \{z_{k,m}\}_{m=1}^{\infty}. \tag{2.3.27}$$

In the above sequences, the front four are monotonically increasing, and the last four are monotonically decreasing. So there exists a sufficiently large constant $M > 10$, and when $m > M$, we have

$$W_{k,m}=\frac{b+\gamma X_{k,m-1}+\gamma Z_{k,m-1}}{b+\mu+\gamma} + \zeta_{8m-7}, \quad X_{k,m} = \frac{b(1-w_{k,m-1})+\delta Z_{k,m-1}}{\beta k \underline{\theta}_{m-1}+\mu} + \zeta_{8m-6},$$
$$Y_{k,m} = \frac{\beta k X_{k,m-1}\overline{\theta}_{m-1}}{\mu+\gamma+\sigma} + \zeta_{8m-5}, \quad Z_{k,m} = \frac{\sigma Y_{k,m}}{\mu+\sigma} + \zeta_{8m-4},$$
$$w_{k,m}=\frac{b+\gamma x_{k,m-1}+\gamma z_{k,m-1}}{b+\mu+\gamma} - 2\zeta_{8m-3}, \quad x_{k,m}=\frac{b(1-W_{k,m-1})+\delta z_{k,m-1}}{\beta k\overline{\theta}_{m-1}+\mu} - 2\zeta_{8m-2},$$
$$y_{k,m}=\frac{\beta k X_{k,m}\theta_{m-1}}{\mu+\gamma+\sigma} - 2\zeta_{8m-1}, \quad z_{k,m}=\frac{\sigma y_{k,m}}{\mu+\delta} - 2\zeta_{8m}, \tag{2.3.28}$$

where $\overline{\theta}_m=\dfrac{\sum\limits_{k} kP(k)Y_{k,m}}{\langle k \rangle}$ and $\underline{\theta}_m=\dfrac{\sum\limits_{k} kP(k)y_{k,m}}{\langle k \rangle}$, $\zeta_m$ satisfies $0 < \zeta_m < \dfrac{1}{m+1}$, and for arbitrarily constant $m > M$ and $t > T_{8m}$, we have

$$W_{k,m} \geq N_{k,m} \geq w_{k,m}, \quad X_{k,m} \geq S_{k,m} \geq x_{k,m},$$
$$Y_{k,m} \geq I_{k,m} \geq y_{k,m}, \quad Z_{k,m} \geq R_{k,m} \geq z_{k,m}.$$

Therefore, all the above sequences have positive limits. Let

$$\lim_{m\to\infty} \left(W_{k,m}, X_{k,m}, Y_{k,m}, Z_{k,m}, w_{k,m}, x_{k,m}, y_{k,m}, z_{k,m}\right) = \left(W_k, X_k, Y_k, Z_k, w_k, x_k, y_k, z_k\right),$$

from (2.3.28), let $m \to \infty$, we have

$$W_k=\frac{b+\gamma X_k+\gamma Z_k}{b+\mu+\gamma}, \quad w_k=\frac{b+\gamma x_k+\gamma z_k}{b+\mu+\gamma},$$
$$X_k=\frac{b(1-w_k)+\delta Z_k}{\beta k\underline{\theta}+\mu}, \quad x_k=\frac{b(1-W_k)+\delta z_k}{\beta k\overline{\theta}+\mu},$$
$$Y_k=\frac{\beta k X_k\overline{\theta}}{\mu+\gamma+\sigma}, \quad y_k=\frac{\beta k X_k\underline{\theta}}{\mu+\gamma+\sigma},$$
$$Z_k=\frac{\sigma Y_k}{\mu+\sigma}, \quad z_k=\frac{\sigma y_k}{\mu+\delta}, \tag{2.3.29}$$

where $\overline{\theta}=\dfrac{\sum\limits_{k} kP(k)Y_k}{\langle k \rangle}$ and $\underline{\theta}=\dfrac{\sum\limits_{k} kP(k)y_k}{\langle k \rangle}$.

From (2.3.29), we have $\dfrac{\sum\limits_{k}\beta k^2 P(k)x_k}{\langle k \rangle} = \mu + \gamma + \sigma = \dfrac{\sum\limits_{k}\beta k^2 P(k)x_k}{\langle k \rangle}$, that is $\sum\limits_{k}\beta k^2 P(k)(X_k-x_k) = 0$, so $X_k=x_k$. Then

we also have $Y_k=y_k$, and $Z_k=z_k$ from (2.3.29). Finally, according to the uniqueness of the solution of $f(\theta) = 0$, we have $X_k = S_k^\infty, Y_k = I_k^\infty, Z_k = R_k^\infty$, that is $\lim\limits_{t\to\infty} S_k(t) = S_k^\infty$, $\lim\limits_{t\to\infty} I_k(t) = I_k^\infty$, $\lim\limits_{t\to\infty} R_k(t) = R_k^\infty$. So $E_+\left(S_k^\infty, I_k^\infty, R_k^\infty\right)$ is globally attractive. Combined with Theorem 5, we get $E_+\left(S_k^\infty, I_k^\infty, R_k^\infty\right)$ is global asymptotically stable when $\rho > 1$ from the LaSalle's invariant principle [20]. This completes the proof. □

The mathematical proofs of global stability offer significant biological insights into ecosystem resilience. The global stability of the disease-free equilibrium (when $\rho < 1$ indicates that the ecosystem is resilient enough to eliminate disturbances; regardless of the initial number of disturbed species, the system will inevitably return to a fully healthy state over time. Conversely, the global stability of the endemic equilibrium (when $\rho > 1$) implies that once the disturbance threshold is crossed, the disturbance becomes persistent and self-sustaining. In this scenario, the ecosystem

shifts to a new stable state where a certain fraction of species remains perpetually disturbed, independent of the initial severity of the attack.

### Remark

We acknowledge that the sequence construction method used in Theorem 5 is technically complex. However, due to the high dimensionality and heterogeneous nature of the ecological network model, constructing an explicit global Lyapunov function for the endemic equilibrium remains a significant mathematical challenge. The monotone iterative technique employed here serves as a rigorous and necessary alternative to establish global stability under these conditions.

## 3. Species protection measures

In order to restrain the spread of disturbance in the ecosystem, human beings often choose some species for protection, which will cut off some paths in the process of disturbance propagation. Next, we propose three different species conservation strategies. According to the influence of protection measures on disturbance propagation threshold, we select the optimal species protection strategy.

### 3.1. Uniform immunization

Let $\varphi$ $(0 < \varphi < 1)$ be the proportion of protected species in the ecosystem, and record it as the immune rate. Assuming that all species in the ecosystem are not distinguished, and species are randomly selected for protection, we have

$$
\begin{cases}
\frac{dS_k(t)}{dt} = b\left(1 - S_k(t) - I_k(t) - R_k(t)\right) - \beta\left(1 - \varphi\right) k\theta(t) S_k(t) - \mu S_k(t) + \delta R_k(t), \\
\frac{dI_k(t)}{dt} = \beta\left(1 - \varphi\right) k\theta(t) S_k(t) - \left(\mu + \gamma\right) I_k(t) - \sigma I_k(t), \\
\frac{dR_k(t)}{dt} = \sigma I_k(t) - \mu R_k(t) - \delta R_k(t), \quad k = 1, \cdots, k_{\max}.
\end{cases}
\tag{3.1.1}
$$

The equilibrium solution of (3.1.1) is

$$
\begin{cases}
S_k^\infty = \frac{b(\mu+\delta)(\mu+\gamma+\sigma)}{(b+\mu)(\mu+\delta)(\mu+\gamma+\sigma) + \left[(\mu+\gamma+\sigma+b)(\mu+\delta)+(b-\delta)\sigma\right]\beta(1-\varphi)k\theta^\infty}, \\
I_k^\infty = \frac{b\beta(1-\varphi)k\theta^\infty(\mu+\delta)}{(b+\mu)(\mu+\delta)(\mu+\gamma+\sigma) + \left[(\mu+\gamma+\sigma+b)(\mu+\delta)+(b-\delta)\sigma\right]\beta(1-\varphi)k\theta^\infty}, \\
R_k^\infty = \frac{\sigma b\beta(1-\varphi)k\theta^\infty}{(b+\mu)(\mu+\delta)(\mu+\gamma+\sigma) + \left[(\mu+\gamma+\sigma+b)(\mu+\delta)+(b-\delta)\sigma\right]\beta(1-\varphi)k\theta^\infty}.
\end{cases}
$$

The self-consistent equation is

$$
\theta^\infty = \frac{1}{\langle k \rangle} \sum_k k P(k) I_k^\infty =
$$

$$
\frac{\sum_k k^2 P(k)}{\langle k \rangle} \frac{b\beta\left(1 - \varphi\right) k\theta^\infty\left(\mu + \delta\right)}{\left(b + \mu\right)\left(\mu + \delta\right)\left(\mu + \gamma + \sigma\right) + \left[\left(\mu + \gamma + \sigma + b\right)\left(\mu + \delta\right) + \left(b - \delta\right)\sigma\right]\beta\left(1 - \varphi\right) k\theta^\infty} \triangleq f\left(\theta^\infty\right).
$$

The threshold condition of its internal equilibrium point is

$$
\left.\frac{df\left(\theta^\infty\right)}{d\theta^\infty}\right|_{\theta^\infty = 0} > 1.
$$

Then we can get the basic reproduction number

$$\rho_1 = \frac{b\beta(1-\varphi)}{(b+\mu)(\mu+\gamma+\sigma)} \frac{\langle k^2 \rangle}{\langle k \rangle} = (1-\varphi)\rho < \rho.$$

This means that the protection of species can effectively inhibit the spread of disturbance.

### 3.2. Target immunization

Because of the heterogeneity of species corresponding nodes, some species with higher degree are in a more important position, and protecting these species can cut off more transmission routes. An upper bound $\kappa$ is introduced so that all nodes with degree greater than $\kappa$ are immune. That is to say, the important species with more predatory relationship with other species are protected. The species with degree $k > \kappa$ are protected, the species with degree $k < \kappa$ are not protected, and the species with degree $k = \kappa$ are protected according to the proportion $c$, let $\varphi_k$ is the immune rate

$$\varphi_k = \begin{cases} 1, & k > \kappa, \\ c, & k = \kappa, \\ 0, & k < \kappa, \end{cases}$$

where $0 < c \leq 1$, and $\sum_k \varphi_k P(k) = \overline{\varphi}$, $\overline{\varphi}$ means the average immune rate. Then we have

$$\begin{cases} \frac{dS_k(t)}{dt} = b\left(1 - S_k(t) - I_k(t) - R_k(t)\right) - \beta\left(1 - \varphi_k\right) k\theta(t) S_k(t) - \mu S_k(t) + \delta R_k(t), \\ \frac{dI_k(t)}{dt} = \beta\left(1 - \varphi_k\right) k\theta(t) S_k(t) - (\mu + \gamma) I_k(t) - \sigma I_k(t), \\ \frac{dR_k(t)}{dt} = \sigma I_k(t) - \mu R_k(t) - \delta R_k(t), \quad k = 1, \cdots, k_{\max}. \end{cases} \tag{3.2.1}$$

The equilibrium solution of (3.2.1) is

$$\begin{cases} S_k^\infty = \frac{b(\mu+\delta)(\mu+\gamma+\sigma)}{(b+\mu)(\mu+\delta)(\mu+\gamma+\sigma)+[(\mu+\gamma+\sigma+b)(\mu+\delta)+(b-\delta)\sigma]\beta(1-\varphi_k)k\theta^\infty}, \\ I_k^\infty = \frac{b\beta(1-\varphi_k)k\theta^\infty(\mu+\delta)}{(b+\mu)(\mu+\delta)(\mu+\gamma+\sigma)+[(\mu+\gamma+\sigma+b)(\mu+\delta)+(b-\delta)\sigma]\beta(1-\varphi_k)k\theta^\infty}, \\ R_k^\infty = \frac{\sigma b\beta(1-\varphi_k)k\theta^\infty}{(b+\mu)(\mu+\delta)(\mu+\gamma+\sigma)+[(\mu+\gamma+\sigma+b)(\mu+\delta)+(b-\delta)\sigma]\beta(1-\varphi_k)k\theta^\infty}. \end{cases}$$

The self-consistent equation is

$$\theta^\infty = \frac{1}{\langle k \rangle} \sum_k kP(k) I_k^\infty =$$

$$\frac{\sum_k k^2 P(k)}{\langle k \rangle} \frac{b\beta\left(1 - \varphi_k\right) k\theta^\infty (\mu+\delta)}{(b+\mu)(\mu+\delta)(\mu+\gamma+\sigma)+\left[(\mu+\gamma+\sigma+b)(\mu+\delta)+(b-\delta)\sigma\right]\beta\left(1-\varphi_k\right)k\theta^\infty} \triangleq f\left(\theta^\infty\right).$$

The threshold condition of its internal equilibrium point is

$$\left. \frac{df\left(\theta^\infty\right)}{d\theta^\infty} \right|_{\theta^\infty = 0} > 1.$$

Then we can get the basic reproduction number

$$\rho_2 = \frac{b\beta\left(\langle k^2\rangle - \langle k^2\varphi_k\rangle\right)}{\langle k\rangle\left(b+\mu\right)\left(\mu+\gamma+\sigma\right)},$$

where $\langle k^2\varphi_k\rangle = \overline{\varphi}\langle k^2\rangle + \varphi'$, $\varphi' = \langle(\varphi_k - \overline{\varphi})(k^2 - \langle k^2\rangle)\rangle$ is the covariance of $\varphi_k$ with $k^2$.

By the similar methods discussed in the targeted immunization program [20], we have $\rho_2 < \frac{1-\overline{\varphi}}{1-\varphi}\rho_1$, let $\varphi = \overline{\varphi}$, so $\rho_2 < \rho_1$, this means that target immunization is more effective than uniform immunization under the same average immunization.

### 3.3. Active immunization

Select a disturbed species node, and protect the species whose neighbor node is larger than $\kappa$, we have

$$\begin{cases} \frac{dS_k(t)}{dt} = b\left(1 - S_k(t) - I_k(t) - R_k(t)\right) - \beta k\theta(t)S_k(t) - \mu S_k(t) + \delta R_k(t), \\ \frac{dI_k(t)}{dt} = \beta k\theta(t)S_k(t) - (\mu+\gamma)I_k(t) - (\sigma+\overline{\varphi}_k)I_k(t), \\ \frac{dR_k(t)}{dt} = (\sigma+\overline{\varphi}_k)I_k(t) - \mu R_k(t) - \delta R_k(t), \quad k = 1, \cdots, k_{\max}. \end{cases} \tag{3.3.1}$$

Where $\overline{\varphi}_k = \frac{1}{\langle k\rangle}\sum_k kP(k)\varphi_k$, and $\varphi_k$ has the same concept as $\varphi_k$ in Section 3.2.

The equilibrium solution of (3.3.1) is

$$\begin{cases} S_k^\infty = \frac{b(\mu+\delta)(\mu+\gamma+\sigma+\overline{\varphi}_k)}{(b+\mu)(\mu+\delta)(\mu+\gamma+\sigma+\overline{\varphi}_k)+\left[(\mu+\gamma+\sigma+\overline{\varphi}_k+b)(\mu+\delta)+(b-\delta)(\sigma+\overline{\varphi}_k)\right]\beta k\theta^\infty}, \\ I_k^\infty = \frac{b\beta k\theta^\infty(\mu+\delta)}{(b+\mu)(\mu+\delta)(\mu+\gamma+\sigma+\overline{\varphi}_k)+\left[(\mu+\gamma+\sigma+\overline{\varphi}_k+b)(\mu+\delta)+(b-\delta)(\sigma+\overline{\varphi}_k)\right]\beta k\theta^\infty}, \\ R_k^\infty = \frac{(\sigma+\overline{\varphi}_k)b\beta k\theta^\infty}{(b+\mu)(\mu+\delta)(\mu+\gamma+\sigma+\overline{\varphi}_k)+\left[(\mu+\gamma+\sigma+\overline{\varphi}_k+b)(\mu+\delta)+(b-\delta)(\sigma+\overline{\varphi}_k)\right]\beta k\theta^\infty}. \end{cases}$$

The self-consistent equation is

$$\theta^\infty = \frac{\sum_k k^2 P(k)}{\langle k\rangle}\frac{b\beta k\theta^\infty(\mu+\delta)}{(b+\mu)(\mu+\delta)(\mu+\gamma+\sigma+\overline{\varphi}_k)+\left[(\mu+\gamma+\sigma+\overline{\varphi}_k+b)(\mu+\delta)+(b-\delta)(\sigma+\overline{\varphi}_k)\right]\beta k\theta^\infty} \triangleq f(\theta^\infty).$$

The threshold condition of its internal equilibrium point is

$$\left.\frac{df(\theta^\infty)}{d\theta^\infty}\right|_{\theta^\infty=0} > 1.$$

Then we can get the basic reproduction number $\rho_3 = \frac{b\beta\left(\langle k^2\rangle - \langle\frac{k^2}{\mu+\gamma+\sigma+\overline{\varphi}_k}\rangle\right)}{\langle k\rangle(b+\mu)}$, so, $\rho_3 < \rho_2$.

### Remark

1. To sum up, we have $\rho_3 < \rho_2 < \rho_1 < \rho$. This means that active immunization is the most effective. Therefore, when choosing the protected species, human beings can find the relatively important species in the predators or prey of the disturbed species, and take protective measures to them, which can more inhibit the spread of disturbance.

2. Practical Implementation. While the theoretical model relies on knowing the real-time state of species, implementation in real ecosystems where observability is limited can be adapted in two ways:

(1) Sentinel Species Monitoring: Conservation efforts can focus on monitoring 'indicator species' that are sensitive to disturbance. Once an indicator species shows signs of instability, protective measures are triggered for its trophic neighbors.

(2) Protection of Keystone Species: Since Active Immunization preferentially targets large-degree neighbors, protecting high-degree 'Keystone Species' (similar to the Targeted Immunization strategy in Section (3.2) serves as the most effective static proxy when real-time state observation is not feasible.

## 4.Numerical simulations

In this section, we study the disturbance propagation process of 85 species in a pine forest in Otago, New Zealand, to study the influence of various parameters on the disturbance propagation process, and prove the stability of disease-free equilibrium and endemic equilibrium. The food web data used in this paper comes from the Interaction Web Database (http://www.ecologia.ib.usp.br/iwdb/). First, we give the food web topology of this ecosystem in Fig 2.

Each point in Fig 2 represents a species, and the line between the two points indicates that there is a predator-prey relationship between the two species. The network visualization is generated using the Fruchterman-Reingold force-directed layout algorithm to reveal structural clustering. The size of each node is proportional to its degree, visually highlighting the high-degree 'hub' species (Keystone Species) that play a critical role in facilitating disturbance propagation. This structural heterogeneity underpins the effectiveness of the Targeted and Active Immunization strategies discussed in Section 3.

Table 1 shows the species represented by all serial numbers:

Now, we calculate the proportion of each degree, and then fit the degree distribution function as shown in Fig 3.

Taking degree as abscissa and degree distribution as ordinate, we can obtain the degree distribution $P(k) = \frac{a}{k}$, $a = 0.2599 \, (0.2331, 0.2868)$, let $a = 0.26$, we have $P(k) = \frac{0.26}{k}$. And then we calculate that $k_{max} = 31$, $\langle k \rangle = 5.36$, $\langle k^2 \rangle = 62.66$. From Fig 4, R-square is 0.9024, it is close to 1, this shows that the fitting is very accurate.

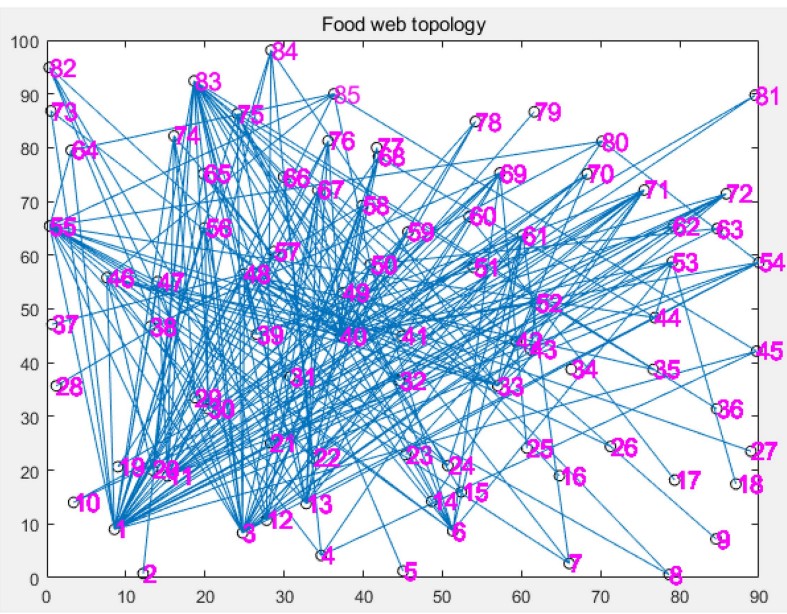

**Fig 2. The topology map of the food web.**

**Table 1. Species name.**

| Number | Species | Number | Species | Number | Species | Number | Species | Number | Species |
|---|---|---|---|---|---|---|---|---|---|
| 1 | Unidentified detritus | 18 | Epithemia sorex | 35 | Nitzschia dissipata | 52 | Cricotopus sp I | 69 | Oligo Lumbri pink |
| 2 | Terrestrial invertebrates | 19 | Eunotia pectinalis | 36 | Nitzschia dubia | 53 | Cricotopus sp II | 70 | Oligo skinny |
| 3 | Plant materials | 20 | Fragilaria vaucheriae | 37 | Nitzschia linearis | 54 | Cristaperla | 71 | Paracalliope fluviatalus |
| 4 | Meiofauna | 21 | Frustulia rhomboides | 38 | Pinnularia spp. | 55 | Deleatidium | 72 | Paracalliope pale |
| 5 | Achnanthes inflata | 22 | Gomphoneis herculeana | 39 | Rhoicosphenia curvata | 56 | Eriopterini | 73 | Paralimnoph-ila skuseii |
| 6 | Achnanthes lanceolata | 23 | Gomphonema accuminatum | 40 | Staurostratum | 57 | Eye forward chiron | 74 | Paucispinigera approximata |
| 7 | Achnanthes linearis | 24 | Gomphonema angustatum | 41 | Surirella elegans | 58 | Helicopsyche | 75 | Polypedellum |
| 8 | Achnanthes minutissima | 25 | Gomphonema intricatum | 42 | Synedra ulna | 59 | Hudsonema aliena | 76 | Polypedel-lum II |
| 9 | Ankitodesmus sp. | 26 | Gomphonema parvulum | 43 | Tabellaria flocculosa | 60 | Hudsonema amabilis | 77 | Polyplectropus |
| 10 | Batrachosper-mum | 27 | Gomphonema sp. III | 44 | Ulothrix | 61 | Hydora nitida (ad) | 78 | Scirtid |
| 11 | Blue-green algae | 28 | Gomphonema sp. unk | 45 | Ameletopsis perscitus | 62 | Hydora nitida (l) | 79 | Sphaerid |
| 12 | Calothrix | 29 | Gomphonema truncatum | 46 | Aoteapsyche | 63 | Hydrobiosella stenocerca | 80 | Stenoperla prasinia |
| 13 | Cocconeis placentula | 30 | Green algae | 47 | Austroperla cyrene | 64 | Hydrobiosis parumbripennis | 81 | Stictocladius |
| 14 | Cymbella kappi | 31 | Gyrosigma | 48 | Austrosimulium | 65 | Mischoderus | 82 | Zelandobius |
| 15 | Cymbella kappii | 32 | Melosira varians | 49 | Chironomini | 66 | Nannochorista phillpotti | 83 | Zelandoperla |
| 16 | Cymbella mulleri | 33 | Navicula avenacea | 50 | Coloburiscus | 67 | Neozephlebia scita | 84 | Zelandotipula |
| 17 | Diatoma heimale | 34 | Navicula rhynocephala | 51 | Costachorema | 68 | Oligo II | 85 | Anguilla dieffenbachii |

Although the simulation is based on the specific fit $P(k) = \frac{0.26}{k}$ which exhibits a high R-square value of 0.9024, it is worth noting that the theoretical derivations in this paper (e.g., the basic reproduction number $\rho$) depend on the statistical moments ($\langle k \rangle$ and $\langle k^2 \rangle$) rather than the specific functional form of the degree distribution. Therefore, the qualitative conclusions regarding stability and disturbance propagation remain robust across different network topologies.

Before presenting the dynamic results, we clarify the data sources and parameter choices used in the simulations:(1) Network Topology: While we provided a power-law fit ($P(k) \approx 0.26/k$) in Fig 3 to describe the network's heterogeneity, the dynamic simulations below utilize the exact empirical adjacency matrix of the 85-species food web. Thus, the results reflect the real topological structure and are independent of the statistical fit of the degree distribution. (2) Dynamical Parameters: The kinetic parameters ($\beta, \sigma, \delta, \mu, \gamma, b$) are selected theoretically to illustrate the system's behavior in different stability regimes. As empirical data for abstract 'disturbance transmission rates' is unavailable, we chose specific values to satisfy the threshold conditions derived in Equation (2.2.12): one set to simulate the outbreak scenario ($\rho > 1$) and another for the extinction scenario ($\rho < 1$). Sensitivity analyses for these parameters are provided in Figs 7–9 to demonstrate the robustness of the conclusions.

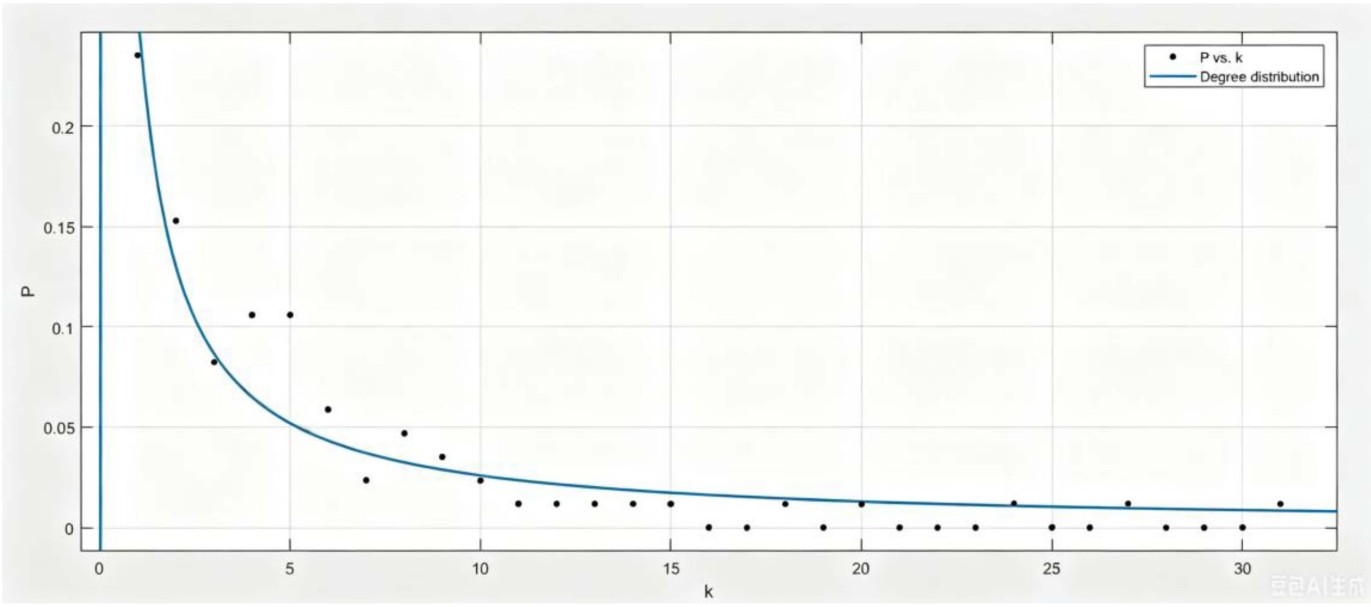

**Fig 3. Degree distribution.**

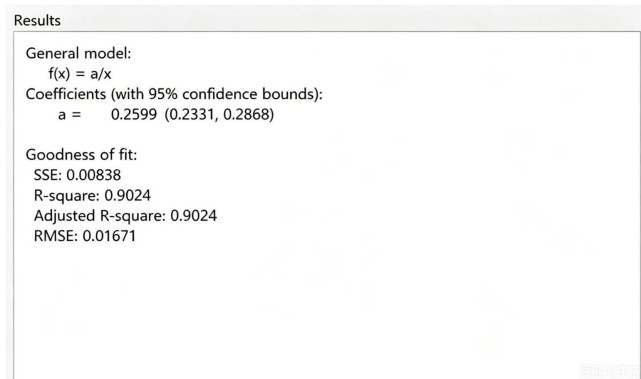

**Fig 4. Results.**

Now, we prove the stability of the equilibrium solution and the process of the solution changing with the corresponding coefficients. It should be noted that while the topological structure of the food web is based on empirical data, the dynamical parameters $\beta, \sigma, \delta, \mu, \gamma, b$ in the following simulations are selected theoretically to illustrate the dynamical behavior and stability properties of the model. To assess the robustness of our results, we performed sensitivity analyses by systematically varying the key parameters, $\beta$, $\sigma$ and $\delta$ as shown in Figs 7–9, respectively. First, we give the initial condition $\mu = b = \gamma = 0.1$, To investigate the deterministic evolution of the system, we solved the differential equations (2.1.1) numerically. The variables $I_k(t)$ presented in Figs 5–9 represent the density of disturbed species within the specific degree class $k$. We initialized the system with a uniform disturbance background: $I_k(0) = 0.1, R_k(0) = 0$, we have $N_k(t) = \frac{b}{b+\mu} = 0.5$. It is worth noting that according to the global asymptotic stability proven in Theorem 2 and Theorem 5,

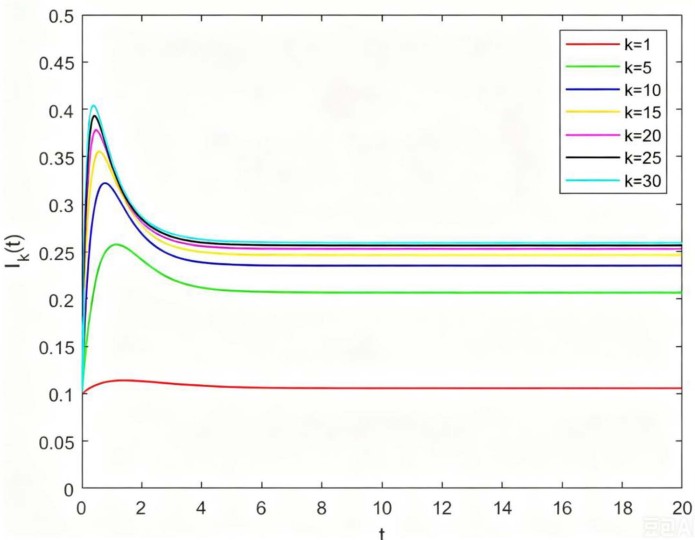

**Fig 5. The relations between $I_k(t)$ and $t$ with different degrees when $\rho > 1$.**

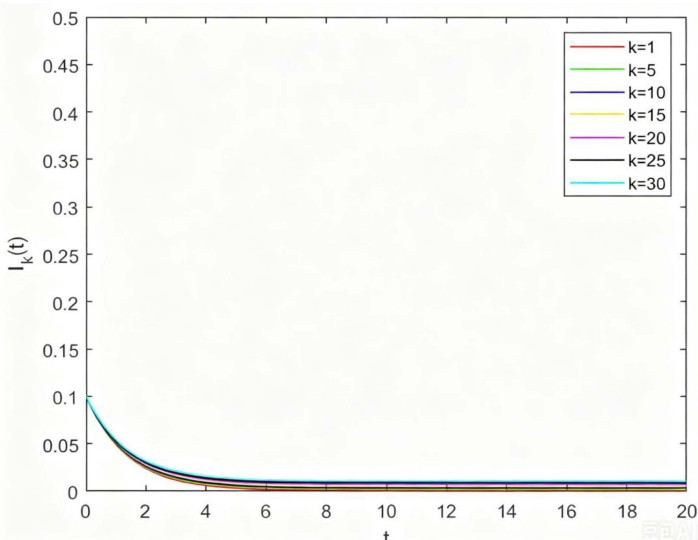

**Fig 6. The relations between $I_k(t)$ and $t$ with different degrees when $\rho < 1$.**

the final equilibrium state is independent of these specific initial conditions. For sparse or localized disturbances (e.g., $I_k(0) \ll 0.1$), the system will converge to the same steady state, albeit potentially requiring a longer transient period.

From Fig 5, let $\beta = 0.8, \delta = 0.5, \sigma = 0.5$, then we have $\rho > 1$, as time $t$ goes on, $I_k(t)$ will first increase and then decrease, and finally stabilize to a state greater than 0. This shows that the endemic equilibrium tends to be stable when the basic reproduction number is greater than 1. At the same time, we see that for different k, the final stable state of $I_k(t)$

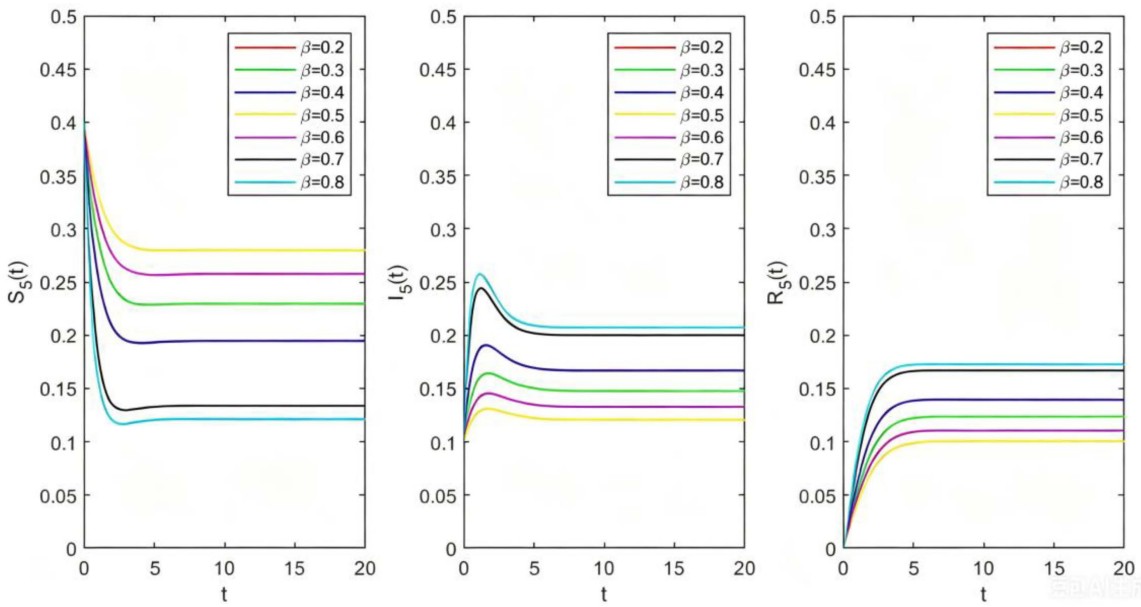

**Fig 7. The relations between $S_k(t)$, $I_k(t)$, $R_k(t)$ and $t$ with different $\beta$.**

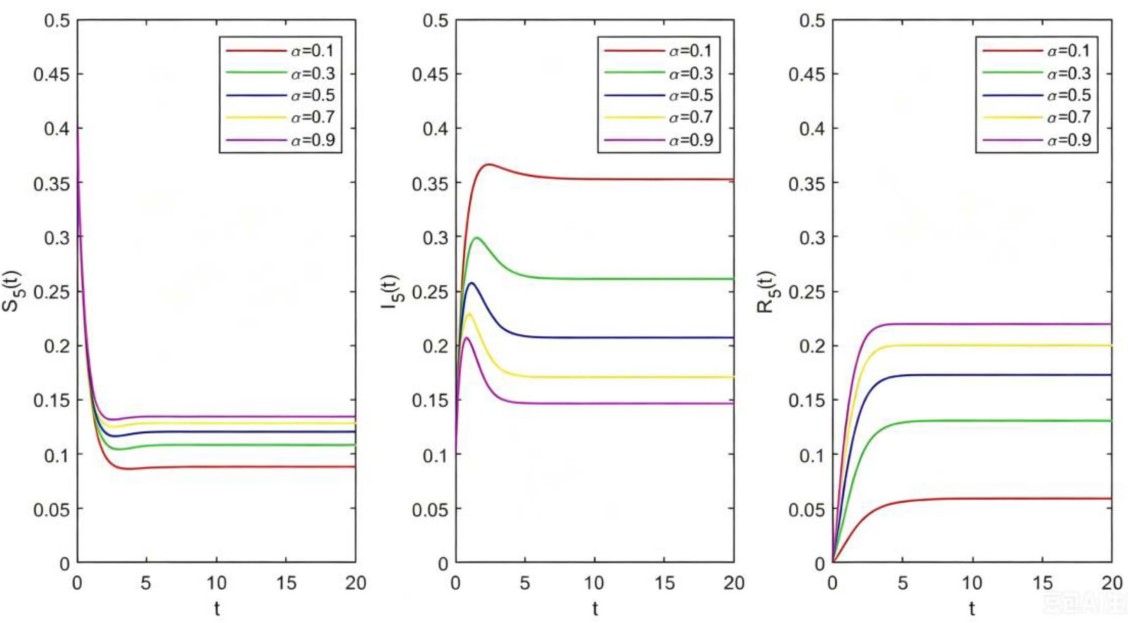

**Fig 8. The relations between $S_k(t)$, $I_k(t)$, $R_k(t)$ and $t$ with different $\sigma$.**

are also different, and $I_k(t)$ increase with the increase of k. This is because that the greater the node degree means that the species has more channels to receive disturbance, so the nodes with higher node degree are more vulnerable to disturbance.

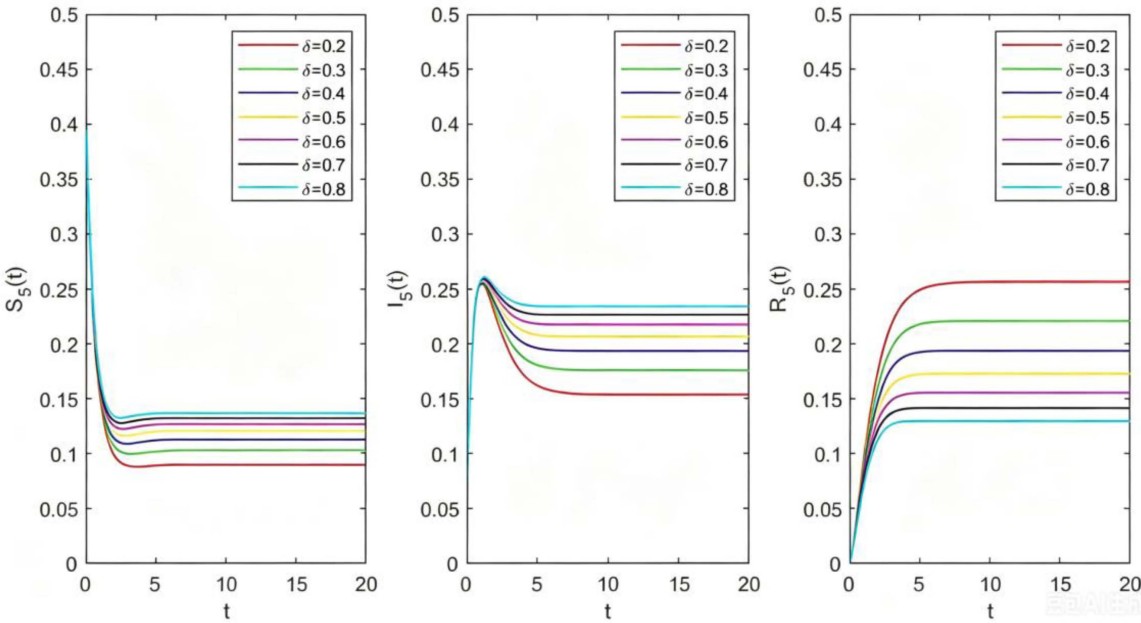

**Fig 9. The relations between $S_k(t)$, $I_k(t)$, $R_k(t)$ and $t$ with different $\delta$.**

From Fig 6, let $\beta = 0.05, \delta = 0.5, \sigma = 0.5$, we can get $\rho < 1$, as time $t$ goes on, $I_k(t)$ will gradually decrease and eventually stabilize to zero. This shows that the disease-free equilibrium tends to be stable when the basic reproduction number is less than 1.

From Fig 7, we can observe that with the increase of the propagation rate $\beta$, the density of disturbed species increases. To further verify the epidemic threshold, we analyzed the relationship between the final stationary density $I(\infty)$ and $\beta$. The results reveal a continuous phase transition (transcritical bifurcation) at the critical value $\beta_c \approx 0.05$ (corresponding to $\rho = 1$). When $\beta < \beta_c$, the system stabilizes at $I(\infty) = 0$; as $\beta$ crosses $\beta_c$ continuously bifurcates from zero and grows monotonically. This numerical behavior is in strict agreement with the theoretical threshold derived in Equation (2.2.13), confirming that $\beta_c$ acts as the precise tipping point for the ecosystem's transition from a healthy to a disturbed state let $k = 5, \delta = 0.5, \sigma = 0.5$ and $\beta = 0.2, 0.3, 0.4, 0.5, 0.6, 0.7, 0.8$, we get the image of $S_k(t), I_k(t), R_k(t)$ changing with t for different $\beta$. We can see that the larger the disturbance propagation probability $\beta$, the easier the disturbance propagates in the system, which will increase the number of disturbed species in the final stable state.

From Fig 8, let $k = 5, \beta = 0.8, \delta = 0.5$ and $\sigma = 0.1, 0.3, 0.5, 0.7, 0.9$, we get the image of $S_k(t), I_k(t), R_k(t)$ changing with t for different $\sigma$. We can see that the greater the probability of species recovery $\sigma$, the more difficult the disturbance is to spread in the system, which will reduce the number of disturbed species in the final stable state.

From Fig 9, let $k = 5, \beta = 0.8, \sigma = 0.5$ and $\delta = 0.2, 0.3, 0.4, 0.5, 0.6, 0.7, 0.8$, we get the image of $S_k(t), I_k(t), R_k(t)$ changing with t for different $\delta$. It indicates the greater the probability that the recovered species will become sensitive species, the easier the disturbance propagates in the system, which will increase the number of disturbed species in the final stable state.

We address the robustness of our stability conclusions with respect to the network structure. A key question is whether the results hold if the specific wiring of the Otago food web is altered. Based on our theoretical derivation of the basic reproduction number $\rho$ (equation 2.2.12), the threshold for disturbance outbreaks depends exclusively on the statistical moments of the degree distribution ($\langle k^2 \rangle / \langle k \rangle$). Randomizing the network topology while preserving the degree distribution preserves these moments. Therefore, the stability criteria derived in this paper are structurally robust and generally

applicable to any network sharing the same heterogeneous degree statistics, regardless of the specific microscopic arrangement of links.

To explicitly verify the effectiveness of the three immunization strategies, we compared the temporal evolution of the disturbance density $I(t)$ under the constraint of a fixed 'immunization cost' (i.e., the same total number of protected species). We set the immunization density $\overline{\varphi} = 0.2$ for all strategies. The simulation results confirm our theoretical prediction based on the basic reproduction numbers ($\rho_3 < \rho_2 < \rho_1$): the Active Immunization strategy leads to the lowest final endemic equilibrium ($I^*$), followed by Targeted Immunization, while Uniform Immunization results in the highest infection level. This demonstrates that for a fixed conservation budget, Active Immunization is the most efficient strategy for suppressing disturbance propagation.

## 5. Conclusion

The propagation dynamics of ecological disturbance based on infectious disease model is an important direction in the field of ecological stability. Combined with complex network theory, this paper studies the propagation dynamics model of disturbance on ecological network. Since the propagation of disturbance is not only related to the topological structure of food web, but also affected by the correlation coefficient of disturbance propagation, this paper focuses on the following aspects:

(1)  This paper studies the dynamic model of disturbance propagation with immigration and emigration in ecological network, and gives the threshold condition of whether the disturbance can break out. We find that when the basic reproduction number is less than 1, the disturbance will not break out; when the basic reproduction number is greater than 1, the disturbance will break out and persist in the ecosystem.

(2) The size of the basic reproduction number is not only related to the topological structure of the network, but also related to the propagation probability of disturbance and the species recovery probability. The increase of disturbance propagation probability and the decrease of species recovery probability will lead to the increase of basic reproduction number, which makes disturbance easier to break out in the ecosystem.

(3) This paper studies the stability of equilibrium solution. We obtain that the disease-free equilibrium is globally stable when the basic reproduction number is less than 1. When the basic reproduction number is greater than 1, the endemic equilibrium is globally stable. At the same time, the conclusion is verified by the actual food web data simulation.

(4) In this paper, human protection of species is regarded as species immunity, and the effects of several different protection measures are discussed according to the change of disturbance propagation threshold. We get that the active immune strategy is the most effective protection measure, that is, the large and medium-sized adjacent nodes of the disturbed species should be protected. This kind of protection is the most effective. However, practical implementation faces significant constraints: (i) Economic Constraints: High-degree species (targets of these strategies) often require disproportionately high financial resources to protect (e.g., large territories for top predators), and 'Active Immunization' incurs additional costs for real-time monitoring. (ii) Ecological Constraints: Achieving perfect 'immunity' is rarely biologically feasible, and intensive intervention may alter natural ecological functions. Therefore, our results should be interpreted as a strategic guide for prioritizing limited conservation resources: identifying the high-cost nodes that yield the highest returns in global ecosystem stability.

Limitations and Future Work:

(1) It should be noted that our current model simplifies the ecosystem into an unweighted network, assuming a uniform propagation probability ($\beta$) for all predator-prey links. In reality, ecological interactions exhibit significant heterogeneity

 

in strength and energy flow, which can differentially impact disturbance propagation. For instance, strong interactions may accelerate instability spread, while weak interactions are often cited as stabilizing mechanisms in food webs. Future iterations of this model will incorporate weighted networks to account for variable interaction strengths and energy fluxes, thereby providing a more granular understanding of how specific pathways drive ecosystem dynamics.

(2) Our model currently employs a uniform propagation probability ($\beta$) for all links, acting as a mean-field approximation. This assumption may limit biological realism by obscuring the stabilizing role of 'weak interactions' or the amplifying effect of 'strong interactions' found in real food webs. Consequently, the model might overestimate disturbance spread in networks dominated by weak links. To address this, the model can be extended by replacing the scalar $\beta$ with a weighted interaction matrix $\beta_{ij}$, where propagation probabilities are proportional to empirical interaction strengths or energy fluxes. While this extension complicates analytical derivation, future work will utilize numerical methods to explore how heterogeneous interaction weights influence the basic reproduction number and global stability.

(3) Finally, the model utilizes the 'empty-lattice' approximation, assuming that the network topology remains static during the disturbance process. This implies that nodes represent persistent ecological niches that are repopulated rather than being structurally removed. We acknowledge that this assumption holds for disturbances acting on a fast times-cale. However, in the case of Keystone Species loss or permanent extinction, the network topology itself would alter, potentially leading to secondary extinctions. Such structural dynamics are beyond the scope of the current SIRS framework but represent a vital avenue for future research combining percolation theory with dynamic modeling.

## Author contributions

**Data curation:** Bingbing Qian.

**Formal analysis:** Bingbing Qian, Xinyue Wang.

**Methodology:** Xinyue Wang, Yimin Li.

**Software:** Bingbing Qian, Yimin Li.

**Supervision:** Jing Hua.

**Writing – original draft:** Jing Hua.

**Writing – review & editing:** Bingbing Qian.

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
