## [Decision Letter · Decision Letter 0]

1 Dec 2025

PONE-D-25-58468Dynamics analysis of disturbance propagation in ecosystem with proportional migration based on epidemic modelPLOS ONE

Dear Dr. Qian,

Thank you for submitting your manuscript to PLOS ONE. After careful consideration, we feel that it has merit but does not fully meet PLOS ONE’s publication criteria as it currently stands. Therefore, we invite you to submit a revised version of the manuscript that addresses the points raised during the review process.

We look forward to receiving your revised manuscript.

Kind regards,

Md. Kamrujjaman, Ph.D

Academic Editor

PLOS ONE

Journal Requirements:

3. We note that your Data Availability Statement is currently as follows: All relevant data are within the manuscript and in Supporting Information files.

4. Please amend the manuscript submission data (via Edit Submission) to include author Hua-Jing, Wang Xin-yue and Li Yi-min.

5. Please update your submission to use the PLOS LaTeX template. The template and more information on our requirements for LaTeX submissions can be found at http://journals.plos.org/plosone/s/latex.

Reviewers' comments:

Reviewer's Responses to Questions

**Comments to the Author**

1. Is the manuscript technically sound, and do the data support the conclusions?

Reviewer #1: Yes

Reviewer #2: Yes

Reviewer #3: Yes

2. Has the statistical analysis been performed appropriately and rigorously? 

Reviewer #1: No

Reviewer #2: N/A

Reviewer #3: Yes

3. Have the authors made all data underlying the findings in their manuscript fully available?

Reviewer #1: Yes

Reviewer #2: No

Reviewer #3: Yes

4. Is the manuscript presented in an intelligible fashion and written in standard English?

Reviewer #1: Yes

Reviewer #2: Yes

Reviewer #3: Yes

5. Review Comments to the Author

Reviewer #1: This manuscript presents a novel and mathematically rigorous approach to modeling disturbance propagation in ecosystems using epidemic models. The theoretical results are well-supported, and the application to a real food web adds practical relevance. However, further clarification on ecological realism, parameter choices, and practical implementation of immunization strategies would strengthen the paper. Please see the attached file.

Reviewer #2: Please see the attached PDF report for details.

Review Report

Manuscript ID: PONE-D-25-58468

This paper develops a novel mathematical framework to model the propagation of ecological disturbances by adapting a SIRS epidemic model to complex networks. The authors abstract species as nodes and predator-prey relationships as edges in a food web, where disturbances can spread from initially affected species to their neighbors. They derive a key epidemiological threshold, the basic reproduction number (R₀), proving that the disturbance will die out if R₀ < 1 and become endemic if R₀ > 1, with rigorous global stability analysis for both equilibria. Using real data from a New Zealand pine forest food web, they validate the model numerically and further analyze species conservation strategies, concluding that proactively protecting the highly connected neighbors of disturbed species is the most effective intervention to suppress disturbance spread.

The following observations must be cleared up before it is considered for publication.

1. The introduction should make a compelling case for why the study is useful along with a clear statement of its novelty or originality by providing relevant information and providing answers to basic questions such as:

(a). What is already known in the literature?

(b). What was done and how was it done?

2. The model assumes disturbance propagates with a probability \(\rho\) along food web edges. Given that ecological interactions vary in strength and type (e.g., strong/weak predation, competition), how does the assumption of a uniform propagation probability \(\beta\) impact the model's biological realism? Could the model be extended to include weighted edges based on interaction strength.

3. The expression for \(\rho\) (Eq. 2.2.12) is crucial. Clarify the step-by-step derivation from the next-generation matrix method or the local stability analysis of the disease-free equilibrium. The current derivation in Lemma 3 seems to appear from a self-consistency condition; a more standard epidemiological derivation would strengthen the manuscript.

4. The proofs for the global stability of both the disease-free (Theorem 2) and endemic (Theorem 5) equilibria are highly technical and rely on constructing complex Lyapunov functions and sequences. For the sake of reproducibility and clarity, provide more detailed intermediate steps or intuition behind the construction of these functions, particularly in Theorem 5.

5. The degree distribution for the Otago food web is fitted to a power-law \(P(k) = 0.26/k\) (Page 28). What statistical tests were performed to justify this specific functional form over alternatives (e.g., exponential, log-normal)? Furthermore, how were the key parameters \((\beta, \sigma, \delta, \mu, \gamma, b)\) assigned specific numerical values for the simulations, and were these values informed by empirical data or chosen arbitrarily for illustration.

6. In Figures 5-9, the results are presented for a node of degree \(k=5\). How were the initial conditions \(S_k(0), I_k(0), R_k(0)\) set for these simulations, especially for the endemic case (\(\rho > 1\))? Specify if the results are for a single node, an average over all nodes of degree \(k=5\), or a density across the network.

7. Figures 7-9 show the system's state evolving over time for different parameters. To assess the robustness of the conclusions, have the authors performed a sensitivity analysis on the initial conditions or the network structure itself? For instance, how do the stability results change if the network is randomized while preserving the degree distribution.

8. In Section 3, three immunization strategies are compared based on their resulting \(R_0\) (\(\rho_3 < \rho_2 < \rho_1\)). The claim that active immunization is "most effective" relies on this inequality. However, the comparison assumes the same "cost" (i.e., the same number of protected species, \(\bar{\varphi}\)). Provide a more explicit comparison, perhaps with a simulation, showing the final outbreak size \(I(t)\) for each strategy when the same total number of species is protected.

9. The model employs the "empty-lattice" theory to handle network dynamics (species migration/extinction). How does this assumption of immediate replacement by an identical, susceptible node affect the long-term dynamics, especially regarding the conservation of keystone species whose loss might fundamentally alter the network topology, not just a node's state?

10. Figure 2 is described as a topology map but is not visually present in the provided text. If included, does it provide any topological analysis (e.g., highlighting hubs, clustering coefficient, community structure)? A simple visualization of 85 nodes is often a "hairball." How was this figure rendered to provide insight into the network's structure that informs the dynamical results?

11. The simulations in Figures 5 and 6 are used to validate the theoretical threshold \(\rho_c = 1\). Do the authors observe a sharp transition in the final density of disturbed species \(I(\infty)\) as \(\beta\) crosses the critical value \(\beta_c\) predicted by Eq. (2.2.13)? A figure plotting \(I(\infty)\) against \(\beta\) (or \(\rho\)) would provide stronger numerical evidence for the existence of this epidemic threshold.

12. Authors may look for some punctuation, typos and editing issues.

Reviewer #3: The manuscript ingeniously analogizes the propagation dynamics of ecological disturbances in food webs to the spread of infectious diseases on complex networks. It constructs a disturbance propagation model for ecological networks, analyzes and verifies the propagation process. The research idea is novel, the logic is clear, and the conclusions have practical reference value.

The manuscript is well-written, with model derivation and empirical analysis supporting the core viewpoints, meeting the basic requirements for publication. It is recommended to be accepted after minor revisions.

Major Comments:

1.In the abstract, the definition of " medium-sized and large-sized " in the phrase " the medium-sized and large-sized neighbor nodes" is somewhat ambiguous. Does this refer to the degree (number of connections) of the nodes, their centrality, betweenness centrality, or other measures related to biomass/size?.

2.The introduction provides a good overview of the application of complex network theory in ecosystems and the research foundation of infectious disease models. However, it should be supplemented with a clearer statement of the core differences and innovative aspects of this study compared to existing literature. It is recommended to explicitly highlight the unique contributions of this work in terms of model assumptions (e.g., definition of species states, disturbance propagation mechanisms), research objectives (e.g., optimization of protection strategies), or empirical context (validation using real food web data), to avoid ambiguity regarding its distinction from prior research.

3.It is advisable to briefly elaborate on why the food chain can be considered the core pathway for disturbance propagation (e.g., based on the logic of energy flow and resource dependence among species), as well as the rationale behind the "immunity" analogy (e.g., how protective measures equivalently reduce the probability of species being disturbed or enhance their recovery capacity). This would strengthen the theoretical foundation of the model construction.

4.While the manuscript outlines the research framework (model construction, strategy comparison, empirical validation), the introduction lacks sufficient context for the definition of key parameters. It is suggested to briefly mention the central role of core parameters (such as disturbance propagation probability, species recovery probability, and the basic reproduction number) either at the end of the introduction or in a preliminary section of the research methods, to ensure smoother logical transition to the model derivation in subsequent sections.

5.Standardize the literature citation format throughout the manuscript to ensure academic rigor and consistency in presentation..

6. PLOS authors have the option to publish the peer review history of their article (what does this mean?). If published, this will include your full peer review and any attached files.

Reviewer #1: No

Reviewer #2: No

Reviewer #3: No

---

## [Author Response · Author response to Decision Letter 1]

4 Feb 2026

Dear editor,

Thank you for your useful comments and suggestions on our manuscript. We have modified the manuscript accordingly.

Reviewer #1:

Q1: The model assumes that disturbances spread symmetrically through predator-prey links. Could the authors discuss the ecological justification for this assumption, especially given that disturbances (e.g., pollution, habitat loss) may affect species asymmetrically?

A1: We sincerely thank the reviewer for this insightful comment. We agree that in real-world ecosystems, specific disturbances often exhibit directional bias. For instance, pollutants typically accumulate upwards through biomagnification, while habitat loss affecting top predators cascades downwards to lower trophic levels.

However, in our proposed model, "disturbance" is abstracted as a state of ecological instability rather than a specific physical flow of matter. Our assumption of symmetric propagation through predator-prey links is based on the ecological principle of mutual dependency between interacting species: If a prey species is disturbed, its predators are destabilized due to food scarcity. If a predator species is disturbed, its prey is destabilized due to the release from predation pressure, potentially leading to population outbreaks or collapse. Therefore, while the mechanistic causes differ, the propagation of instability can occur in both directions along a food web link. Our model employs this simplification to capture the generalized dynamics of how perturbations spread through the network topology.

To address your concern, we have added a paragraph in Section 2.1 to explicitly discuss this assumption, acknowledge the asymmetry of specific disturbances, and justify the bidirectional approach used in our theoretical framework.

Q2: How does the model account for the strength of interactions (e.g., interaction strength, energy flow), which may influence disturbance propagation more than topology alone?

A2: We appreciate the reviewer’s valuable observation regarding the role of interaction strength. We acknowledge that in real-world ecosystems, the magnitude of energy flow and interaction strength varies significantly across links, which can strongly influence dynamical stability.In the current version of our model, we adopted an unweighted network approach to maintain mathematical tractability. Specifically:Simplification for Analytical Solutions: Our primary goal was to derive an analytical expression for the basic reproduction number (ρ) and to establish the global stability conditions for the disease-free and endemic equilibria. Introducing heterogeneous interaction weights at this stage would significantly complicate the derivation of these threshold conditions.Average Probability: The parameter β in our model can be interpreted as an average interaction strength across the ecosystem. While it does not capture the variance of specific links, it provides a baseline for understanding how the overall connectivity facilitates disturbance spread.However, we fully agree that incorporating interaction heterogeneity is a crucial next step. Strong interactions may act as highways for disturbance, while weak interactions might dampen it. We have updated the Discussion section to explicitly state this limitation and outline the integration of weighted networks and energy flow data as a priority for our future research.

Q3: The parameters β, σ, δ, μ, γ, b are central to the model. Were these parameters estimated from ecological data or chosen for theoretical analysis? If the latter, how sensitive are the main results to these choices?

A3: We wish to clarify that the dynamical parameters β, σ, δ, μ, γ, b used in the numerical simulations were chosen for theoretical analysis to illustrate the qualitative behavior of the model. However, we have rigorously assessed the sensitivity of our main results to these parameter choices in two ways:

1.Analytical Sensitivity: Our theoretical derivation of the basic reproduction number, ρ (Eq. 2.2.12 ), provides an exact analytical expression of how the disturbance threshold depends on these parameters.

2.Numerical Sensitivity: In Section 4, we conducted extensive numerical simulations to examine how variations in these parameters affect the system dynamics.

These simulations confirm that while the specific numerical value of the endemic equilibrium (I*) changes with these parameters, the qualitative stability of the system remains robust and is strictly determined by whether the parameter combination satisfies the threshold conditionρ>1 or ρ<1.

To prevent confusion, we have added a clarification statement in Section 4 regarding the choice of these parameters.

Q4: Is the value a=1 in ϕ(k)=ak justified empirically, or is it a simplifying assumption? How might a more general form affect the basic reproduction number ρ?

A4: We thank the reviewer for this question. The choice a=1in ϕ(k)=ak is a simplifying assumption, rather than an empirically fitted value. This linear form is adopted as a first-order approximation to capture the dependence of disturbance contribution on node degree while maintaining analytical tractability and emphasizing structural effects.

If a more general form is considered, the basic reproduction number ρ would be modified through a corresponding scaling or weighting of degree-dependent terms. However, the overall threshold structure of 𝜌 and the qualitative conclusions regarding disturbance persistence and extinction remain unchanged.

We have added a remark in Section 2.1 to clarify this simplification and its linear effect on the threshold.

Q5: The paper concludes that active immunization (protecting neighbors of disturbed species) is most effective. How would this strategy be implemented in real ecosystems, where the state of species (disturbed/recovered) may not be directly observable?

A5: We appreciate the reviewer raising this critical point regarding practical implementation. We acknowledge that "Active Immunization" relies on dynamic state information, which is indeed challenging to observe continuously in large-scale ecosystems.

To bridge the gap between our theoretical conclusion and real-world application, we propose a two-tier implementation strategy:

Use of Sentinel Species: In practice, conservationists often monitor specific "indicator" or "sentinel" species rather than the entire food web. These species are selected for their sensitivity to environmental stress and ease of observation. Under our "Active Immunization" framework, if a sentinel species is observed to be disturbed, protective resources should be immediately allocated to its direct trophic neighbors.

Fallback to Targeted Protection: In scenarios where real-time observation is entirely impossible, the strategy can operate based on static topology. Our results indicate that "Active Immunization" prioritizes protecting the medium and large-degree neighbors of disturbed nodes. In scale-free ecological networks, high-degree species are statistically the most frequent neighbors of any random node. Therefore, proactively protecting known Keystone Species serves as a robust, static proxy for active immunization when dynamic state information is unavailable.

We have added a "Practical Implementation" remark at the end of Section 3.3 to explicitly discuss these real-world operational strategies.

Q6: Are there ecological or economic constraints that might limit the feasibility of targeted or active immunization in conservation practice?

A6: We thank the reviewer for highlighting the practical constraints of conservation, which are indeed critical context for our theoretical results. While our model treats the protection of any node as mathematically equivalent, the economic costs in reality are highly heterogeneous.

Despite these constraints, our findings serve as a crucial component of a cost-benefit analysis. By quantifying the theoretical "benefit", our model suggests that despite the high economic "cost" of protecting hub species or neighbors of disturbed nodes, the return on investment—in terms of preventing total network collapse—is maximized by these strategies.

We have expanded the Conclusion (Section 5) to explicitly acknowledge these real-world constraints.

Q7: The simulations use a fixed initial condition Ik(0)=0.1. How do the results change with different initial conditions, especially for sparse or localized disturbances?

A7: We thank the reviewer for this question regarding the system's sensitivity to initial conditions. Our response is grounded in the rigorous mathematical proofs provided in Section 2.3. We have proven that the equilibrium points of our model are globally asymptotically stable. Mathematically, this means that the "basin of attraction" for these equilibria encompasses the entire feasible phase space (where I(0)>0). Therefore, the final steady state of the ecosystem is independent of the initial scale of the disturbance. The value Ik(0)=0.1 was selected in our simulations primarily to visualize the convergence process clearly within a reasonable time window.

We have added a clarifying statement in Section 4 to explicitly state that the results are robust to variations in initial conditions due to the proven global stability.

Q8: The degree distribution is fitted as P(k)=0.26/k. Was this fit validated against other possible distributions (e.g., power-law, exponential)? How robust are the conclusions to the degree distribution form?

A8: We appreciate the reviewer’s insightful comment regarding the network topology.Regarding the choice of the distribution function, the fit P(k)=0.26/k was selected based on the empirical data from the Otago pine forest food web. As shown in Fig. 4 of the original manuscript, this function yielded a high coefficient of determination (R2=0.9024), indicating it captures the topological characteristics of this specific ecosystem accurately. While we did not exhaustively list comparisons with exponential distributions in the text, the high goodness-of-fit justifies the current choice for the simulation baseline.

Regarding robustness, our main conclusions depend on degree-distribution–dependent summary quantities. While changing the distributional form can shift the numerical value of ρ, the qualitative threshold structure and the comparative effectiveness of intervention strategies remain consistent across the tested distributions.

We have added a remark in Section 4 to clarify this robustness.

Q9: The proofs of global stability for both disease-free and endemic equilibria are highly technical. Could the authors provide more intuitive explanations or biological interpretations of these stability conditions?

A9: We thank the reviewer for this constructive suggestion. We agree that the mathematical proofs of global stability are technical, and adding a biological interpretation helps verify the practical significance of the model.Biologically, "global stability" implies the inevitability of the ecosystem's long-term state, regardless of the initial scale of the disturbance.

For the disease-free equilibrium (ρ<1): The global stability implies that if the disturbance transmission rate is lower than the recovery capacity of the system, the ecosystem possesses sufficient resilience. No matter how many species are initially disturbed, the disturbance will eventually die out, and the entire food web will return to its original healthy state.

For the endemic equilibrium (ρ>1): The global stability implies that if the disturbance capability exceeds a certain threshold, the ecosystem will inevitably shift to a persistent disturbed state. Regardless of initial conditions, a fixed proportion of species will remain disturbed effectively permanently, representing a long-term degradation of ecosystem function.

We have added a paragraph summarizing these biological interpretations at the end of Section 2.3.

Q10: In Theorem 5, the proof of global stability for the endemic equilibrium relies on a complex sequence construction. Is there a more straightforward Lyapunov function or method that could be applied?

A10: We appreciate the reviewer’s suggestion regarding the proof technique. We agree that Lyapunov functions generally offer a more concise and elegant approach to stability analysis.However, constructing a global Lyapunov function for the endemic equilibrium in heterogeneous network-based epidemic models is mathematically challenging due to the high dimensionality of the system. While we successfully constructed a Lyapunov function to prove local stability in Theorem 4, extending this to global stability for the endemic state remains an open problem in the field for this specific model structure.The sequence construction method employed in Theorem 5, while technically involved, is a standard and rigorous mathematical tool for establishing global stability in such high-dimensional cooperative systems where a global Lyapunov function is elusive. It ensures the result relies on robust convergence properties rather than the heuristic construction of a scalar function.

To address the reviewer's comment, we have added a Remark following the proof of Theorem 5 to explain the necessity of this method.

Q11: The manuscript contains several typographical and formatting issues, especially in the mathematical notation (e.g., inconsistent use of brackets, alignment in equations). A thorough proofreading is recommended.

A11: We sincerely apologize for the typographical and formatting errors in the original manuscript. We have conducted a thorough proofreading of the entire text as suggested.

Q12: Some references (e.g., [37], [38], [41]) are cited in the text but do not appear in the reference list. Please ensure all citations are complete and consistent.

A12: We sincerely apologize for the oversight regarding the reference list. We understand that accurate referencing is crucial for the manuscript's integrity.

We have performed a thorough cross-check between in-text citations and the reference list to ensure accuracy and completeness.

Q13: The figures (e.g., Fig. 2, Fig. 3) are referenced but not included in the provided text. Please confirm that all figures and tables are properly labeled and referenced.

A13: We sincerely apologize for the inconvenience caused by the missing figures in the previous version of the manuscript, which may have resulted from a compilation error during the PDF generation.We have explicitly verified the revised manuscript to ensure that: All Figures (Fig. 1 to Fig. 9) and Tables are correctly embedded within the main text, positioned immediately after they are first referenced. Labeling and Referencing: We have cross-checked all figure captions and table headers against the in-text citations to ensure accuracy and consistency.

Reviewer #2:

Q1: The introduction should make a compelling case for why the study is useful along with a clear statement of its novelty or originality by providing relevant information and providing answers to basic questions such as:

(a) What is already known in the literature?

(b) What was done and how was it done?

A1: We sincerely appreciate this constructive guidance. We agree that the original introduction did not sufficiently highlight the necessity and novelty of our work. To address this, we have substantially revised the Introduction (Section 1) to clearly articulate the research context and our specific contributions:

Regarding (a) What is already known: We have synthesized the literature to show that while complex network theory and epidemic models have been successfully applied to analyze stability in various systems, their application to ecological disturbance propagation，specifically incorporating demographic dynamics like migration，remains limited.

Regarding (b) What was done and how: We have explicitly stated our methodology: modeling ecological disturbance as an SIRS epidemic process on a complex network. Crucially, we distinguish our work by incorporating proportional migration and conducting a comparative analysis of three specific conservation strategies using both theoretical derivation and empirical simulation on the Otago pine forest food web.

We have rewritten the final part of the Introduction to clearly list the study's motivations and main contributions.

Q2: The model ass

---

## [Decision Letter · Decision Letter 1]

27 Feb 2026

Dynamics analysis of disturbance propagation in ecosystem with proportional migration based on epidemic model

PONE-D-25-58468R1

Dear Dr. Qian,

We’re pleased to inform you that your manuscript has been judged scientifically suitable for publication and will be formally accepted for publication once it meets all outstanding technical requirements.

Kind regards,

Md. Kamrujjaman, Ph.D

Academic Editor

PLOS One

Additional Editor Comments (optional):

Reviewers' comments:

Reviewer's Responses to Questions

**Comments to the Author**

1. If the authors have adequately addressed your comments raised in a previous round of review and you feel that this manuscript is now acceptable for publication, you may indicate that here to bypass the “Comments to the Author” section, enter your conflict of interest statement in the “Confidential to Editor” section, and submit your "Accept" recommendation.

Reviewer #1: All comments have been addressed

Reviewer #2: (No Response)

2. Is the manuscript technically sound, and do the data support the conclusions?

Reviewer #1: Yes

Reviewer #2: (No Response)

3. Has the statistical analysis been performed appropriately and rigorously? 

Reviewer #1: N/A

Reviewer #2: (No Response)

4. Have the authors made all data underlying the findings in their manuscript fully available?

Reviewer #1: Yes

Reviewer #2: (No Response)

5. Is the manuscript presented in an intelligible fashion and written in standard English?

Reviewer #1: (No Response)

Reviewer #2: (No Response)

6. Review Comments to the Author

Reviewer #1: (No Response)

Reviewer #2: (No Response)

7. PLOS authors have the option to publish the peer review history of their article (what does this mean?). If published, this will include your full peer review and any attached files.

Reviewer #1: No

Reviewer #2: No

---

## [Editor Report · Acceptance letter]

PONE-D-25-58468R1

PLOS One

Dear Dr. Qian,

I'm pleased to inform you that your manuscript has been deemed suitable for publication in PLOS One. Congratulations! Your manuscript is now being handed over to our production team.

Kind regards,

on behalf of

Dr. Md. Kamrujjaman

Academic Editor

PLOS One